# Three distinct strategies lead to programmable aliphatic C−H oxidation in bicyclomycin biosynthesis

Lian Wu[1,8], Jun-Bin He [2,8], Wanqing Wei [3,8], Hai-Xue Pan [2,4], Xin Wang [5], Sheng Yang [1] ✉, Yong Liang [5,6] ✉, Gong-Li Tang [2,4] ✉ & Jiahai Zhou [7] ✉

The C−H bond functionalization has been widely used in chemical synthesis over the past decade. However, regio- and stereoselectivity still remain a significant challenge, especially for inert aliphatic C−H bonds. Here we report the mechanism of three Fe(II)/α-ketoglutarate-dependent dioxygenases in bicyclomycin synthesis, which depicts the natural tactic to sequentially hydroxylate specific C−H bonds of similar substrates (cyclodipeptides). Molecular basis by crystallographic studies, computational simulations, and site-directed mutagenesis reveals the exquisite arrangement of three enzymes using mutually orthogonal strategies to realize three different regio-selectivities. Moreover, this programmable selective hydroxylation can be extended to other cyclodipeptides. This evidence not only provides a naturally occurring showcase corresponding to the widely used methods in chemical catalysis but also expands the toolbox of biocatalysts to address the regioselective functionalization of C−H bonds.

Carbon-hydrogen (C−H) functionalization has been established as a powerful strategy to construct diverse natural products and organic molecules in synthetic chemistry[1–5]. However, achieving site- and stereo-selective functionalization remains a formidable challenge, especially among aliphatic ($sp^3$) C−H bonds (the dissociation energy of C−H bond shown in Supplementary Fig. 1)[6–11]. Two chemical procedures are developed to address these challenges. One relies on the substrate which is based on inherent reactivity or contains a directing group, or the reaction is conducted intramolecularly. The other relies on chemical catalysts, which use ligand architecture or spatial orientation to control selectivity, often involving transition metals (Fig. 1a)[6,9,12–15]. In most cases, the site-selectivity is predictable according to the reaction mechanism, but it is still hard to achieve high enantio-selectivity. By contrast, enzymatic catalysis often shows excellent selectivity by fitting a specific C−H bond of substrate in the active site[10,13,14,16–24]. However, it is difficult to predict this selectivity without precise geometrical information from enzyme-substrate complex structure and detailed energetic calculations. Thus, studying the mechanism of enzymatic C−H activation to provide a predictable selectivity model is highly desirable.

Numerous enzymes have been identified in various natural product biosynthetic pathways, which perform different C−H functionalization reactions such as hydroxylation, desaturation, halogenation, and alkylation. Enzymatic C−H hydroxylation that exhibits extraordinary selectivity is often catalyzed by several metalloenzymes, including non-heme Fe(II)/α-ketoglutarate-dependent dioxygenase

[1]Key Laboratory of Synthetic Biology, Chinese Academy of Sciences (CAS) Center for Excellence in Molecular Plant Sciences, University of CAS, Shanghai 200032, China. [2]State Key Laboratory of Chemical Biology, Shanghai Institute of Organic Chemistry, University of CAS, Shanghai 200032, China. [3]School of Biotechnology and Key Laboratory of Industrial Biotechnology of Ministry of Education, Jiangnan University, Wuxi 214122, China. [4]School of Chemistry and Material Sciences, Hangzhou Institute for Advanced Study, University of CAS, Hangzhou 310024, China. [5]Henan-Macquarie University Joint Centre for Biomedical Innovation, School of Life Sciences, Henan University, Kaifeng 475004, China. [6]State Key Laboratory of Coordination Chemistry, Jiangsu Key Laboratory of Advanced Organic Materials, Chemistry and Biomedicine Innovation Center, School of Chemistry and Chemical Engineering, Nanjing University, Nanjing 210023, China. [7]State Key Laboratory of Microbial Technology, Nanjing Normal University, Nanjing 210023, China. [8]These authors contributed equally: Lian Wu, Jun-Bin He, Wanqing Wei. ✉e-mail: syang@sibs.ac.cn; yongliang@nju.edu.cn; gltang@sioc.ac.cn; jiahai@nnu.edu.cn

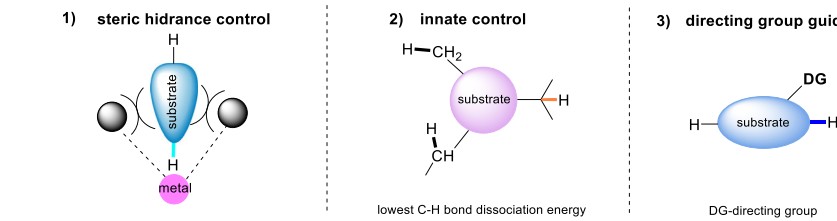

**Fig. 1 | The strategies to catalyze C−H activation. a**, The general chemical synthetic approach of C−H activation. **b**, Examples of enzymes to control C−H activation in natural product biosynthesis. 1) Two highly homologous Rieske oxygenases hydroxylate the substrate on different sites by the shape of the active pocket. 2) Two αKGDs (non-heme Fe(II)/α-ketoglutarate-dependent dioxygenase) chlorinate lysine at C-4 and C-5, respectively. 3) Three homologous αKGDs catalyze three different types of C−H bond activation, BCM is the abbreviation of biclomycin.

(αKGD), cytochrome P450 monooxygenase, Rieske oxygenase and copper monooxygenase[7,16,17,19,25–36]. Some of them were well characterized, for example, two Rieske oxygenases SxtT and GxtA, which share 88% sequence identity, catalyze site-selective C−H hydroxylation reactions in paralytic shellfish toxin biosynthesis (Fig. 1b1)[37,38]. Two αKGDs, BesD and HalB, could chlorinate lysine at C-4 and C-5, respectively (Fig. 1b2)[39]. Several αKG-dependent amino acid hydroxylases can hydroxylate the same amino acid at different sites. For example, VioC, OrfP, CmnC, and EFEs catalyze the hydroxylation of L-Arg or its analogs, while P3H and P4H selectively oxidize the C−H bonds of proline[40–49]. The substrates involved in the above reactions are highly functionalized or serve as prerequisites for subsequent oxidation reactions. Chemically, precise spatial and chiral recognition of a directing template now enables the iterative C−H editing of special substrate (quinoline pharmacophores) at any desired site and order[50]. In contrast, the example of homologous enzymes that catalyze C−H hydroxylation with different site-selectivities in the same biosynthetic pathway is extremely rare. Three αKGDs (BcmE/BcmC/BcmG) catalyze site- and stereo-selective hydroxylation of inert C−H bonds in BCM's (bicyclomycin) cyclic dipeptide scaffold (Fig. 1b3), one of the simplest peptide derivatives without additional functional groups[51–55]. Despite extensive prior characterization of the BCM pathway[51–55], the selectivity mechanisms of these enzymes remain unexplored.

In this work, we constructed truncated thoezyme models based on reported spectroscopic, crystallographic and computational data in αKGDs[22,56–65] to investigate the inherent site selectivity of cognate substrates (**1**–**3**). Contrary to expectations, our analysis uncover divergent selectivity profiles between substrates **1** and **3** relative to thoezymatic system. Through integrating crystallographic studies, computational simulations and biochemical experiments, we identify key residues modulating selectivity of three αKGDs. Our hypothesis posits distinct regulatory mechanisms: BcmE employs steric hindrance, BcmC relies on inherent reactivity, while BcmG utilizes a directing group to control C−H hydroxylation. Substrate scope validation confirms this framework, each enzyme exhibits characteristic substrate preferences consistent with their proposed strategies. This study provides a systematic mechanistic basis for understanding how αKGDs (BcmE/BcmC/BcmG) achieve orthogonality in site selectivity during BCM biosynthesis.

## Results

### BcmE and BcmG employ different strategies to control site selectivity compared to BcmC

The mechanism of aliphatic C−H hydroxylation by αKGDs has been investigated thoroughly, with processing via hydrogen abstraction and radical rebound, and the rate-limiting step is the former[22,42,66–72]. To reveal the inherent site selectivity of hydroxylation, we conducted density functional theory (DFT) calculations on the cognate substrates (**1**–**3**) catalyzed by a truncated theozyme model, which contained $Fe^{IV}$-oxo, two methylimidazoles for two histidine residues, two acetate anions in aspartic acid and succinate, as well as one water (Fig. 2a). Given the activation free energy of hydrogen abstraction on the C-2′ (5.1 kcal mol⁻¹) is the lowest for BcmC, we predicted the site selectivity

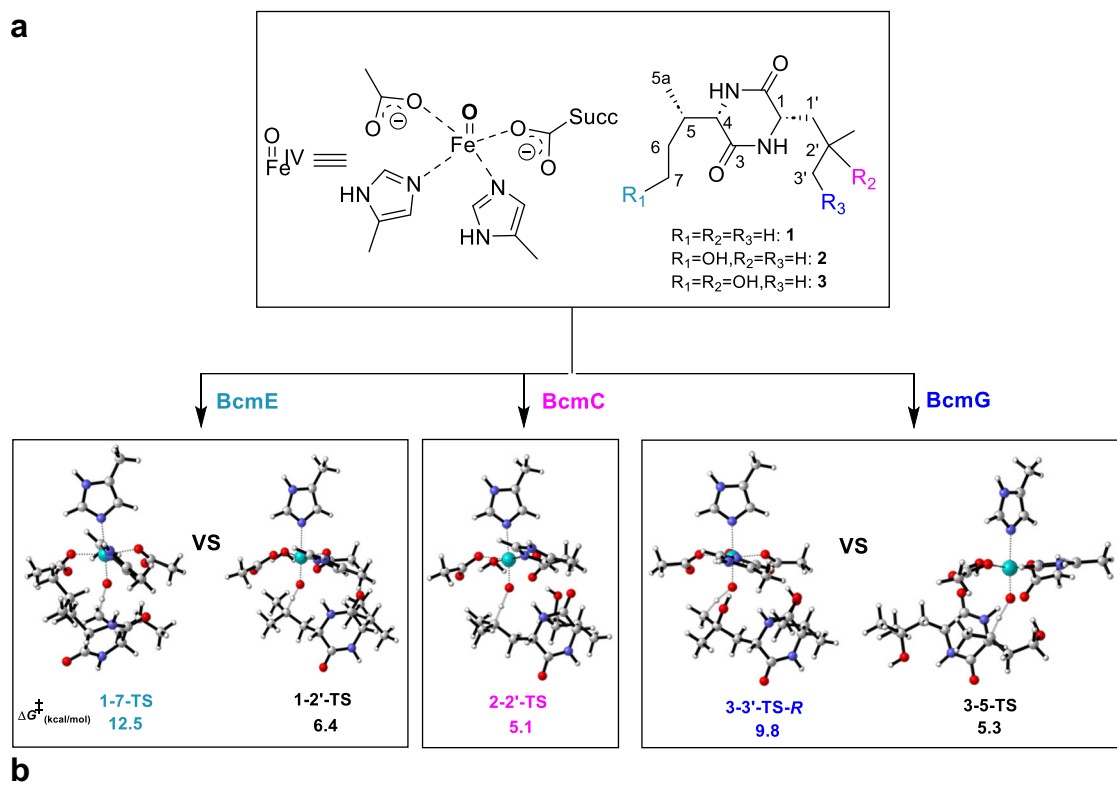

**Fig. 2 | DFT-computed transition states and Gibbs free energies barrier (in kcal mol⁻¹) for the hydrogen abstraction reactions by a truncated catalytic-residue theozyme model. a**, Computations at the CPCM (chlorobenzene)-B3LYP-D3/6-311 + G(2 d,p)[SDD for Fe]//CPCM-B3LYP-D3/6-31 G(d,p)[LanL2DZ for Fe] level of theory and transition state structures (carbon: gray, hydrogen: white, oxygen: red, nitrogen: blue, iron: cyan). **b**, The energy barriers of every C−H bond are listed in the table, and the reaction sites are highlighted.

**b**

| Calculated energy barriers for different types of hydrogen abstraction (kcal/mol) | | | |
|---|---|---|---|
| site | BcmE | BcmC | BcmG |
| 1' | 9.9(11.8) [$R(S)$] | 8.2(12.1) [$R(S)$] | 9.3(11.7) [$S(R)$] |
| 2' | 6.4 | 5.1 | --- (-OH) |
| 3' | 10.8(12.9) [$S(R)$] | 9.4(10.4) [$R(S)$] | 9.8(12.8) [$R(S)$] |
| 5 | 8.7 | 6.9 | 5.3 |
| 5a | 14.4 | 11.9 | 7.9 |
| 6 | 6.6(8.2) [$R,R(R,S)$] | 8.7(10.0) [$R,R(R,S)$] | 9.1(10.8) [$R,S(R,R)$] |
| 7 | 12.5 | --- (-OH) | --- (-OH) |

of BcmC-catalyzed hydroxylation depends primarily on the innate reactivity of substrate. Indeed, BcmC selectively hydroxylates the C-2' position of **2**. However, the lowest energy barrier on the C-2' (6.4 kcal mol⁻¹) is lower than that on the C-7 for BcmE, while the lowest energy barrier on the C-5 (5.3 kcal mol⁻¹) is lower than that on the C-3' for BcmG (Fig. 2b and Supplementary Figs. 2–4). As expected, substrate **1-3**, catalyzed by a theozyme without the protein scaffold, has the highest activity of tertiary carbon, C-2', C-2', and C-5, respectively. In fact, BcmC targets the most reactive position (C-2') but BcmE and BcmG react with sites that are intrinsically less reactive (C-7 not C-2', C-3' not C-5), indicating that the specific enzyme scaffold is another essential factor influencing the regioselectivity of the substrate[39,45,47,61,73–76].

### Structural characterizations of BcmE, BcmC and BcmG

To investigate the effect of the microenvironment of these three enzymes on the site selectivity, we attempted to determine the crystal structures of SsBcmE, SsBcmC, and SsBcmG from *Streptomyces sapporonensis*. Despite the extensive screening, we only obtained the binary or ternary complexes crystals of SsBcmE and SsBcmC, with Fe(II) or/and αKG, but failed for that of SsBcmG. We subsequently selected the homologous enzymes and finally obtained the crystals of the ternary complexes of PaBcmG (from *Pseudomonas aeruginosa*), which also catalyzed the conversion of substrate **3** to **4** (Supplementary Fig. 5). As a result, we determined a collection of X-ray crystal structures including ternary complexes of SsBcmE, SsBcmC, PaBcmG (with Fe(II) or/and αKG) (Supplementary Table 1). These three structures shared the conserved "jelly-roll" core[42,66] and possessed the HXD-$X_n$-H motif[42,77] coordinated with iron (Supplementary Figs. 6a–c). Although these three enzymes share only 35–42% amino acid identities (Supplementary Fig. 7), the overall structures are similar, and r.m.s.d. values are 0.70 Å (SsBcmE vs SsBcmC), 1.07 Å (SsBcmE vs PaBcmG) and 0.87 Å (SsBcmC vs PaBcmG), respectively (Supplementary Fig. 6d). The primary distinction among the three structures lies in the C-terminal helix and the shape of active pocket (Supplementary Fig. 6d). The C-terminal helix is too flexible to be modeled, especially that of SsBcmE. And the B-factor of that is also much higher than that of other parts of structures (Supplementary Fig. 6d). To further determine the

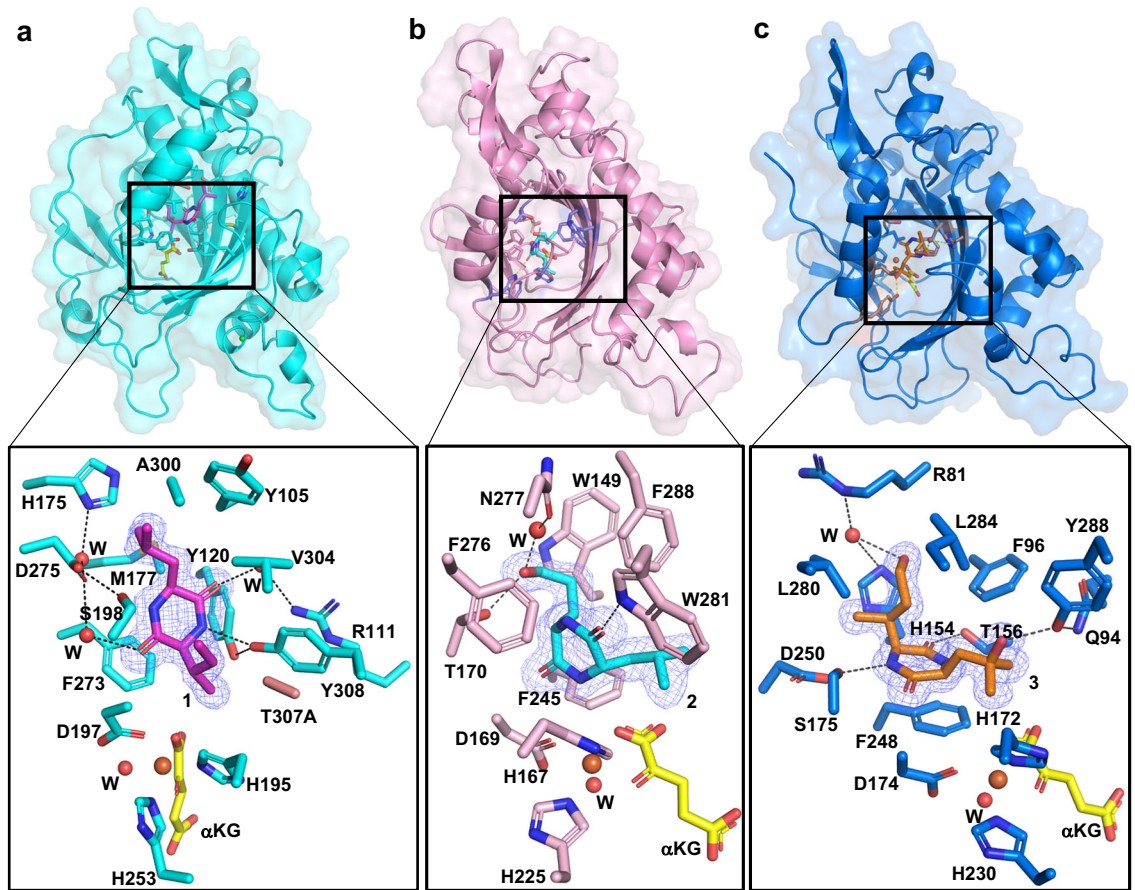

**Fig. 3 | The substrate-bound complex structures reveal detailed active sites of Fe(II)- and αKG- dependent SsBcmE, SoBcmC and PaBcmG. a–c** The active pocket of the complex structures of SsBcmE^T307A•Fe^II•αKG•**1** (cyan), SoBcmC•Fe^II•αKG•**2** (pink) and PaBcmG•Fe^II•αKG•**3** (blue), respectively. The above is the overall cartoon structure and the below is the detailed interactions of

corresponding substrates and residues (iron: orange, nitrogen: blue, oxygen: red) shown as sticks. The omit map (blue mesh) for substrates in active site is contoured to 3.0σ. The carbon atoms of αKG, **1**, **2** and **3** are colored in yellow, magenta, cyan and orange, respectively.

key residues controlling the site selection, we tried to solve the quaternary complex structures of SsBcmE, SsBcmC and PaBcmG with their corresponding substrates. Except for solving the quaternary complex of PaBcmG•Fe^II•αKG•**3**, we did not get the other two complex structures using SsBcmE and SsBcmC.

We tried many ways to determine the quaternary complex structure of SsBcmE•Fe^II•αKG•**1** by various approaches: expressing the protein in distinct expression host, utilizing diverse fusion tags, homologous enzymes and setting crystallization experiments at a range of temperatures with multiple concentration gradients and mutant variants. Eventually, we solved the structure of SsBcmE^T307A•Fe^II•αKG•**1** (PDB ID: 8XHY) (Fig. 3a, Supplementary Fig. 8a and Supplementary Table 1), which could catalyze the hydroxylation at C-5a (vide infra, Supplementary Fig. 9). The carbonyl group of the amide part of the DKP ring is stabilized by a π-π stacking interactions with the aromatic ring F273. Additionally, Y308 forms hydrogen bonds with the NH group of the DKP ring and the hydroxyl group of Y120. Two water molecules mediate the hydrogen interactions between carbonyl groups of DKP and residues R111 and D275. S198 interacts with D275 by a hydrogen bond, too. Another water molecule mediates the hydrogen bonding network between D275 and H175. The 5-methyl group of the DKP ring is oriented towards the iron center, suggesting that hydrogen abstraction is likely to occur preferentially at the C-5a position. This observation is consistent with the activity of SsBcmE^T307A (Supplementary Fig. 9). The presence of a rotatable single bond between C-4 and C-5 brings the possibility of C-7 closer to the iron center, thereby preserving the reactivity observed in the wild-type

SsBcmE (Supplementary Fig. 9). Given that the mutant complex structure did not fully exhibit the characteristics of the wild-type protein complex, we modeled the structure of wild-type SsBcmE based on the mutant quaternary complex structure and docked substrate **1** to the active sites. We found the steric hindrance of T307 could bring C-7 closer to the iron center (Supplementary Figs. 9, 10a). Residues M177, V304, and A300 form the hydrophobic portion of the cavity to accommodate the hydrophobic portion of **1** (Fig. 3a). In our molecular dynamics (MD) snapshot, we observed that specific residues are arranged around substrate **1**, forming steric hindrance (Supplementary Fig. 10a). This arrangement suggests that these residues may play a crucial role in substrate binding and significantly influence the selectivity of hydroxylation. Therefore, we hypothesize that the selectivity of BcmE-catalyzed C − H oxidation stems from the steric effect formed by the catalytic cavity.

Similarly to above, we determined the quaternary complex structure of SoBcmC•Fe^II•αKG•**2** (PDB ID: 8XHQ) (Figs. 3b, Supplementary Fig 8b and Supplementary Table 1) by crystallizing the homologous enzyme SoBcmC from *Streptomyces ossamyceticus*. The DKP ring of substrate **2** is locked in the hydrophobic cavity formed by F245, F276 and F288. Two carbonyl groups of the DKP ring are involved in the formation of direct hydrogen bonds with the imino group of W281 and the water-mediated hydrogen bonds with the amide group of N277, respectively. Steric interaction between W149 and Ile moiety of **2** causes the 7-OH of **2** to move upward, forming a hydrogen bond with T170 at a distance of 2.7 Å. Notably, the distance between C-3' and the iron center (5.0 Å) seems closer than that of C-2' (5.7 Å), which

contradicts the hydroxylation selectivity of BcmC (Fig. 2b). We speculated that the crystal structure only captured one state of **2**, and the single C-1′ − C-2′ bond of **2** is rotatable which could rotate to the state that C-2′ directs to the iron center, facilitating the hydroxylation reaction. This is in agreement with the results of DFT (Supplementary Fig. 3) and MD calculations (Supplementary Fig. 10b).

The quaternary complex structure of PaBcmG•Fe$^{II}$•αKG•**3** (PDB ID: 8XHX) (Fig. 3c, Supplementary Fig. 8c and Supplementary Table 1) reveals that the DKP ring in substrate **3** is securely positioned above F96, L280 and L284 through van der Waals interactions. The DKP ring interacts with F248 via π-π stacking, while its two carbonyl groups form hydrogen bonds with T156 and S175, respectively. The 7-OH of **3** forms water-mediated hydrogen bonds with H154 and R81, whereas the 2′-OH group forms a hydrogen bond with the phenolic hydroxyl group of Y288 which couples with the amine group of Q94. Furthermore, the C-3′ is closer to the iron center, which is consistent with the result that BcmG catalyzed the C-3′ hydroxylation (Supplementary Fig. 10c). Likewise, the comparison between PaBcmG•Fe$^{II}$•αKG and PaBcmG•Fe$^{II}$•αKG•**3** structures revealed that the C-terminal helix may function as a gate or lid for the active pocket[39,78], and the binding of substrate triggers the closure of this gate (Supplementary Fig. 11e, f). Our results suggest that the interactions between PaBcmG residues and substrate **3** control the substrate to be in a specific orientation for hydroxylation, which effectively reverses its inherent site selectivity.

## BcmE, BcmC and BcmG use three different strategies to control the site selectivity of C − H hydroxylation

Based on the structural analyzes above, we employed MD simulations and site-directed mutagenesis experiments to investigate the mechanisms behind the site selectivity and stereoselectivity of C−H hydroxylation by SsBcmE, SoBcmC and PaBcmG. We supposed that these three enzymes controlled the site selectivity by modulating the active conformation to favor the desired site of hydroxylation. According to the DFT-optimized structures of transition states, we defined the active conformation that would achieve site-specific hydroxylation as O − H distance is ≤ 2.8 Å (the sum of van der Waals atomic radii of carbon and oxygen atoms) and angle O − H − C is in the range of 170 ± 15°. We constructed the reactive species configurations of SsBcmE•Fe$^{IV}$-oxo•succinate•**1**, SoBcmC•Fe$^{IV}$-oxo•succinate•**2** and PaBcmG•Fe$^{IV}$-oxo•succinate•**3**, and obtained the corresponding representative MD snapshots (Supplementary Fig. 10d–f).

Among 5000 conformations of **1** in the SsBcmE•Fe$^{IV}$-oxo•succinate•**1** complex, there are only ten active conformations for C-7 hydroxylation, while there are no active conformations for other sites (Supplementary Fig. 10a–d). We hypothesized that a constrained cavity composed of bulky residues Y120, F273, V304, T307, and Y308 forced C-7 in **1** into a pre-reactive conformation through van der Waals interactions. Consistent with our assumption, site-directed mutagenesis showed that Y105A almost lost the enzyme activity, as well as Y120A, M177A, V304A, and Y308A drastically decreased it, indicating the critical role of these residues (Supplementary Fig. 9). Conversely, mutations Y120F, V304T, T307A, and Y308F enhanced total activities (72%−95% total conversion yields) and produced a new C-5a-hydroxylated product **1a** (2%−27% conversion yields, Supplementary Fig. 9). Surprisingly, the mutants T307L (62% conversion yield, 100% selectivity) and Y308M (21% conversion yield, 100% selectivity) completely reverse the site selectivity of C − H hydroxylation, yielding a C-2′-hydroxylated product **1b**, akin to the reaction catalyzed by BcmC. MD simulations (Supplementary Fig. 12c, d) indicated that the larger side chain of T307L increased steric hindrance with substrate **1**, altering its binding and favoring the reaction at the lower energy barrier of C-2′. Given the structural proximity of Y308M to T307L, we hypothesize that Y308M may similarly affect substrate binding. Additionally, the mutation F273A completely terminates the C-7 hydroxylation activity

but exhibits the C-6 hydroxylation to afford **1c** (58% conversion yield, 100% selectivity, Supplementary Fig. 9). These findings suggest that the residues F273, T307 and Y308 in SsBcmE play a crucial role in regulating the site selectivity. We also modeled mutants F273A, T307L, and T307A complexes by using a pre-equilibrated SsBcmE•Fe$^{IV}$-oxo•succinate•**1** complex structure as a template (Supplementary Fig. 12a−c, e). As expected, the dominant active conformations for hydroxylation in the F273A, T307L and T307A systems are C-6, C-2′ and C-5a (shown in the dotted rectangle: 49/5000, 90/5000 and 14/5000, Supplementary Fig. 12b−d, f), respectively, which is consistent with our mutagenesis results and structures, particularly the crystal structure of SsBcmE$^{T307A}$•FeII•αKG•**1**. The active conformations for C-7 in these three mutants account for only 3/5000, 3/5000, and 4/5000, respectively (Supplementary Fig. 12b−d, f), further supporting the idea that steric hindrance mediated by these residues controls site selectivity in SsBcmE-catalyzed hydroxylation.

Together, we propose that BcmE regulates site selectivity by constructing a specific hydrophobic cavity that accommodates the orientation of substrate **1**, employing a steric hindrance control strategy similar to those used in chemical synthesis (Fig. 4a). For instance, mutating the bulky F273 to alanine reduces the steric hindrance on the leucine side of substrate **1**, positioning the preferred reactive site, C-6, closer to the center and facilitating hydroxylation. This observation is consistent with the reactivity predicted in Fig. 2b.

In SoBcmC•Fe$^{IV}$-oxo•succinate•**2** complex, we found that C-2′ hydroxylation is the dominant active conformation (shown in the dotted rectangle: 122/5000, in Supplementary Fig. 10e), which is accordant with the DFT calculation result that the activation free energy of hydrogen abstraction on the C-2′ is the lowest for BcmC-catalyzed hydroxylation reaction. Structural analyzes demonstrate that residues F245, F276 and F288 orient the DKP ring in substrate **2** into the active cavity via hydrophobic interactions. However, the single-site mutations of F245A, F276A and F288A have almost no effect on the activity and selectivity (Supplementary Fig. 13), probably because the cavity of the SoBcmC is too large to affect the substrate binding and orientation (volume of SoBcmC is 252 Å$^3$, but that of SsBcmE and PaBcmG are 139 Å$^3$ and 210 Å$^3$, respectively). The W149A mutation dramatically reduces the activity, owing to W149 may lock the substrate in the active pocket (Fig. 3b). In addition, mutagenesis of T170, N277 and W281, which constituted the hydrogen bond network with substrate **2**, resulted in the reduction of C-2′ hydroxylation activity (19%−75% conversion yields) but performed the C-5 hydroxylation activity (4%−23% conversion yields) to generate product **2a** (Supplementary Fig. 13), which is also the inherently more reactive site with the lowest activation free energy (Fig. 2b). Consequently, these findings suggest that the site selectivity of BcmC-catalyzed hydroxylation mainly depends on the innate selectivity of substrate **2**, in which one C − H bond is inherently more reactive than the other bonds. Such a substrate control strategy (Fig. 4c) is commonly used in chemical synthesis (Fig. 1a2).

In PaBcmG•Fe$^{IV}$-oxo•succinate•**3** complex, the active C-3′ hydroxylation conformation is dominant (shown in the dotted rectangle: 327/5000, Supplementary Fig. 10f). However, the above DFT calculations manifested that the activation free energy of hydrogen abstraction on the C-3′ is the highest in the BcmG theozyme model (Fig. 2b). Combined with the crystal structure analyzes, we speculated that the hydrogen bonding interactions medicated by R281, Q94, H154, T156, S175 and Y288 and the hydrophobic interactions mediated by F96, F248, L280 and L284 stabilize and orient the substrate **3** in a suitable conformation to favor the C-3′ hydroxylation. To investigate our assumptions of the putative key residues, we performed systematic site-directed mutagenesis of PaBcmG (Supplementary Fig. 14). The mutations F96A, T156A, S175A, L280A and F248A substantially decrease the activities, and the mutant L284A completely lost its activity (Supplementary Fig. 14), suggesting these

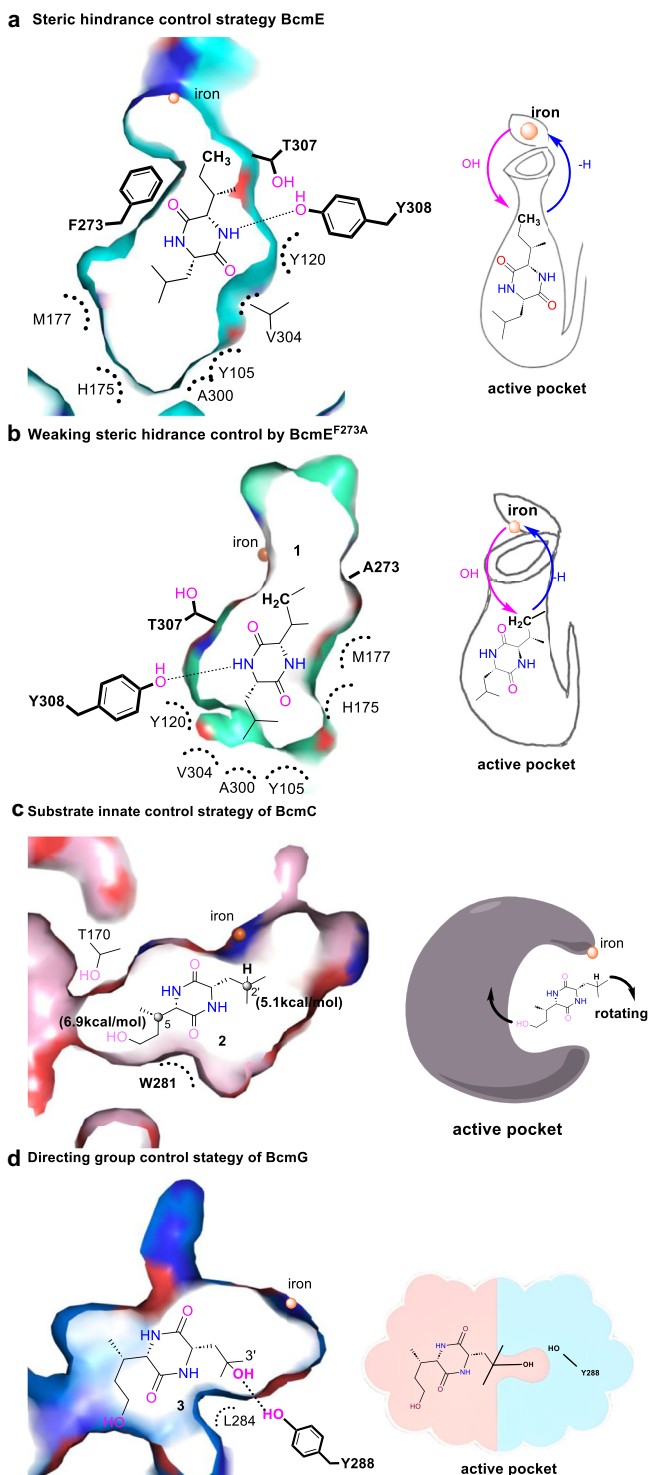

**a** Steric hindrance control strategy BcmE

**b** Weaking steric hidrance control by BcmE^F273A

**c** Substrate innate control strategy of BcmC

**d** Directing group control stategy of BcmG

**Fig. 4 | The proposed site selectivity mechanisms of SsBcmE, SsBcmE^F273A, SoBcmC and PaBcmG. a–d** The corresponding mechanisms (left, cutview of active pockets) and their concept maps (right). In the right panels, we present simplified models of the active pockets to clarify the selectivity mechanisms. It's important to note that these models do not represent the full enzymes, and the iron centers are embedded within the active pockets, not exposed on the enzyme surface.

residues play a crucial role in enzyme activity. Similarly, the H154A and R81A variants reduce the activity, indicating that the water-mediated hydrogen bonds with 7-OH contribute to the substrate binding for hydroxylation. Moreover, the hydrogen bonding interactions between Y288 (also interact with Q94) and 2'-OH drives C-3'

in substrate **3** to adopt a catalytically active conformation for hydroxylation (Supplementary Fig. 14). Expectedly, the activity of Y288A is completely abolished and the activity of Q94A is markedly diminished. More importantly, we found that the substitution of Y288 to F and W reversed the C-2' diastereoselectivity to produce a diastereomeric product **3a** (100% selectivity) and its five-membered derivates **3b/3c** (Supplementary Fig. 14). All these bioactive results further confirmed the role of these key residues. Collectively, we speculate that BcmG performs exquisite control over the site selectivity and stereoselectivity mainly through the directed guidance mediated by the hydrogen bond between Y288 and the 2'-OH of **3**, similar to a directing group control strategy in chemical catalysis (Fig. 4d).

## Mechanism-guided selective C−H hydroxylation of 2,5-diketopiperazines

Based on the above results, we have obtained three enzymes that can hydroxylate the inert C−H bond from CH, CH₂ and CH₃ (Fig. 5a and Supplementary Fig. 15), respectively. The different site selectivity strategies of BcmE/BcmC/BcmG also inspired an investigation of whether these αKGDs would activate C−H bonds of other cyclodipeptides. To this end, several cyclodipeptides condensed from hydrophobic amino acids were used as substrates to test the catalytic selectivity of these three enzymes (Fig. 5b, c). As expected, BcmE only recognized isoleucyl-containing cyclodipeptides including **5, 6**, and **7**, and hydroxylated the C-7 position of the isoleucine moiety of these compounds to yield **5a** (59% conversion yield), **6a** (12% conversion yield) and **7a** (16% conversion yield), indicating that BcmE exhibited high regioselectivity (up to 100%) and substrate specificity toward these unactivated isoleucyl-containing DKP scaffolds (Figs. 5b and Supplementary Fig. 16). BcmC could specifically hydroxylate the tertiary carbons (C-2') of the leucine moiety in leucyl-containing cyclodipeptides (**1** and **8–10**) with high catalytic activity (up to 100% conversion yield) and excellent selectivity (up to 100%) (Figs. 5c and Supplementary Fig. 17). As calculated above (Fig. 2), these results also confirm that the site-selectivity of BcmC-catalyzed hydroxylation depends mainly on the innate reactivity of the substrate. For BcmG, it preferentially catalyzes the hydroxylation of DKP compounds (**2** and **8a–10a**) with a hydroxyl group in vicinal position with high regioselectivity (up to 100%) and activity (Figs. 5c and Supplementary Fig. 18). For example, BcmG was able to hydroxylate the natural intermediate **2** at C-6 to form **2b**, further supporting the importance of the directing group (i.e., the ortho hydroxyl group) for the hydroxylation site-selectivity of BcmG (Figs. 5c and Supplementary Fig. 15 and Supplementary Fig. 18). Depending on the reaction characteristics, we speculated that the C−H hydroxylation of 2,5-diketopiperazines could be programmed by these mutually orthogonal enzymes to obtain various hydroxylated products (Fig. 5c). Collectively, these data further support the proposed regulatory mechanism that BcmE, BcmC, and BcmG precisely tune site-selective C−H hydroxylations by using different strategies (Fig. 4a, c and d). This also implies that the αKGDs BcmE, BcmC, and BcmG diversified the enzyme toolkit, providing excellent biocatalysts for site-selective and stereoselective C−H hydroxylation of 2,5-diketopiperazines.

## Discussion

Activating inert C−H bonds while controlling stereo- and regioselectivity is a significant challenge in chemical synthesis[25,50]. Many catalysts have been developed to tackle this issue, but they typically rely on substrate control, such as the intrinsic reactivity of the substrate (including electronic, steric, and stereoelectronic effects) or pre-installed directing groups[2,10], or the architecture of chemical catalyst. All this procedure involves the installation and removal of the catalyst's directing group or coordinated ligands, which adds steps to experiments[2]. In contrast, natural enzymes employ different

**Fig. 5 | Site-selective and stereoselective enzymatic C–H hydroxylation of 2,5-diketopiperazines. a** SsBcmE-, SsBcmE[F273A]- and SoBcmC-catalyzed site-selective C–H activations. **b** SsBcmE-catalyzed regioselective C–H hydroxylation of 2,5-diketopiperazines. **c** SsBcmE-, SoBcmC-, and PaBcmG-catalyzed the programmable C–H hydroxylation for the synthesis of diverse 2,5-diketopiperazines. [a]The percentage refers to regioselectivity. [b]The percentage refers to diastereoselectivity.

strategies to discriminate between similar C–H bonds and activate them during natural product biosynthesis. Arguably, nature exemplifies the power of these approaches, where a cumulative set of interactions between the functionalities present in the substrate and the active site imbues nature's enzymes with their exquisite selectivity[79].

In this study, we investigated the mechanisms employed by three homologous enzymes, BcmE, BcmC, and BcmG, which catalyze the hydroxylation of **1**–**3** sequentially. We specially investigated the residues in the second coordinate sphere of these three enzymes and found out it was consistent with previous research, that the second coordinate sphere determines the substrate recognition and selectivity[80–83]. Our analysis revealed that these enzymes utilize mutually orthogonal strategies to achieve regio- and stereo-selectivity: steric hindrance control, substrate innate control, and directing group control. Further experiments with expanded set of substrates demonstrated that BcmE can preferentially activate CH$_3$ group, BcmC targets the C–H bond with the lowest bond dissociation energy, and BcmG favors the site near the directing hydroxy group. The active assays of mutants showed that engineered BcmE or BcmG could gain new selectivity against WT.

These findings provide valuable insights into how the αKGDs distinguish closely related substrates and catalyze regioselective hydroxylation in BCM biosynthesis. Moreover, the mutually orthogonal mechanisms of these three enzymes ensure the predictable and programable aliphatic C–H activation to hydroxylate different sites on substrates. This offers a promising approach for future applications in chemical synthesis and the development of novel biocatalysts. Continued advancement of this approach may lead to the ultimate realization of molecular editing: the freedom to modify organic molecules at any site, in any order.

## Methods
### General
Chemicals, biological reagents, media, enzymes and Kits were purchased from standard commercial sources unless otherwise stated.

Compounds **1**–**4**, **1b**, **1 d**, **4a**, and **4b** were previously reported by our laboratory[51,84,85]. Compounds **5**–**10** were purchased from Leon (Nanjing) Biotechnology Co., Ltd. Oligonucleotide primers were synthesized, and DNA was sequenced by TsingKe Biotech Co., Ltd. (Shanghai, China). PCR amplification was performed using a C1000 Touch™ Thermal cycler (Bio-Rad, USA). All the protein quantification was measured using a NanoDrop 2000 spectrophotometer (Thermo Fisher Scientific Inc., USA). High performance liquid chromatography (HPLC) analysis was performed on an Agilent 1260 instrument (Agilent Technologies, Germany). Liquid chromatography coupled with mass spectrometry (LC-MS) analysis was performed on a Thermo Fisher LTQ Fleet mass spectrometer (Thermo Fisher Scientific) equipped with an electrospray ionization (ESI) source. Enzymatic products were isolated by semi-preparative HPLC on a Shimadzu LC-20-AT system (Shimadzu Corporation, Japan). High resolution (HR) ESI-MS data were obtained on a Bruker maXis 4 G instrument. NMR spectra were recorded on Agilent 500/54 Premium Shielded 500 MHz NMR spectrometer and Bruker AscendTM 600 MHz spectrometer with CD$_3$OD as the solvent. Chemical shifts (δ) are given in parts per million (ppm) and coupling constants (*J*) are given in hertz (Hz).

### Protein expression and purification
The glycerol stock (*E. coli* BL21(DE3) strain) harboring plasmid pET-37b-SsbcmE was previously reported by our laboratory[51,84,85]. The gene encoding *SobcmC* from *Streptomyces ossamyceticus* and *PabcmG* from *Pseudomonas aeruginosa* were codon-optimized for heterologous expression in *E. coli* BL21 (DE3) and synthesized by TsingKe Biotech Co., Ltd. (Shanghai, China). The synthesized genes were subsequently cloned into the pET-37b(+) vector (pre-linearized with *Nde I* and *Xho I*). The verified recombinant plasmids were transformed into *E. coli* BL21 (DE3) competent cells (TransGen Biotech, China) for protein expression. A single clony was inoculated into 10 mL Luria-Bertani (LB) medium supplemented with 50 μg mL$^{-1}$ kanamycin at 37 °C overnight. Then, 8 mL of an overnight seed culture was used to inoculate 800 mL of LB media supplemented with 50 μg mL$^{-1}$ kanamycin. Cells were

grown at 37 °C with shaking (220 rpm) until an optical density at 600 nm (OD$_{600}$) reached 0.6–0.8. Protein expression was induced by addition 0.4 mM isopropyl β-D-thiogalactoside (IPTG), followed by incubation at 16 °C for another 20 h with shaking (200 rpm). The cells were harvested by centrifugation at 7,600 g for 5 min at 4 °C, resuspended in 30 mL of pre-cooled lysis buffer (50 mM Tris-HCl, 300 mM sodium chloride, and 10 mM imidazole; pH 7.5), and disrupted by sonication in an ice bath. After centrifugation at 63,000 g for 15 min at 4 °C, the clarified supernatant was incubated with 3 mL of Ni-NTA resin for 1 h at 4 °C with gently shaking. The mixture was then loaded onto a column, and the resin was washed with 30 mL of lysis buffer. The bound recombinant protein was eluted using a stepwise gradient of imidazole (20 mM, 50 mM, and 300 mM) in elution buffers (50 mM Tris-HCl, 300 mM sodium chloride). The purified protein was concentrated and buffer-exchanged into storage buffer (50 mM Tris-HCl, 150 mM sodium chloride; pH 7.5) using an Amicon Ultra-30K (Millipore, USA). Protein purity was confirmed by SDS-PAGE, and protein concentration was determined by measuring absorbance at 280 nm using a Nanodrop 2000 spectrophotometer (Thermo Fisher Scientific). The concentration values were corrected using the theoretical extinction coefficient calculated by the ProtParam tool on the Expasy server (https://web.expasy.org/protparam/). Finally, the purified proteins were flash-frozen in liquid nitrogen and stored at −80 °C for further experiments.

### Enzyme activity assays and HPLC and LC-MS analyzes

Each enzymatic reaction (50-µL scale) consisted of 50 mM Tris-HCl buffer (pH 7.5), 0.6 mM substrate, 2 mM α-Ketoglutaric acid (αKG), 2 mM L-ascorbic acid, 0.1 mM FeSO$_4$·7H$_2$O, and 50 µg of purified enzyme. The reactions were performed at 37 °C for SoBcmC and PaBcmG, and at 16 °C for SsBcmE. The reaction times were 12 h for SsBcmE, 2 h for SoBcmC, and 1 h for PaBcmG. A heat-inactivated enzyme (boiled at 100 °C for 15 min) served as the negative control. After the reactions, 100 µL of pre-cooled methanol was added to quench the reactions, followed by centrifugation at 13,800 g for 30 min. The supernatants were then analyzed by HPLC and LC-MS. HPLC and LC-MS analyzes were performed according to our previous methods[84,85].

### Site-directed mutagenesis and enzyme activity assay of mutants

Site-directed mutants were constructed using a Fast Mutagenesis System kit (TransGen Biotech) with the corresponding primers listed in Supplementary Table 2–4. Protein expression, purification, and enzyme activity assays of all these mutants were performed according to the same protocol described for the wild-type (WT) enzymes.

### Protein expression and purification for crystallization

Protein SsBcmE was expressed in *E. coli* BL21(DE3) in 2×YT medium supplemented with 50 µg mL$^{-1}$ kanamycin at 37 °C. When OD$_{600}$ reached 0.8–1.0, protein expression was induced by 0.4 mM IPTG at 16 °C for 18 h. Cells were harvested by centrifugation at 7600 g for 3 min at 4 °C. The cells were resuspended in pre-cooled buffer A containing 25 mM Tris (pH 8.0), 500 mM NaCl, 5 mM β-mercaptoethanol, and 1 mM phenylmethylsulfonyl fluoride (PMSF), and lysed by French press with a high-pressure homogenizer (500–600 bar). Cell debris was removed by centrifugation at 63,000 g for 15 min at 4 °C. The supernatant was applied to a pre-equilibrated Ni-NTA affinity column (GE Healthcare, USA), sequentially washed with 100 mL buffer A, followed by an additional wash with a buffer consisting of 25 mM Tris (pH 8.0), 500 mM NaCl, 30 mM imidazole, and 5 mM β-mercaptoethanol. The His-tagged protein was eluted using buffer B, which consisted of 25 mM Tris (pH 8.0), 500 mM NaCl, 300 mM imidazole, and 5 mM β-mercaptoethanol. The purified protein was concentrated using an Amicon Ultra-30KDa (Millipore), and subsequently loaded to a pre-equilibrated Hi Load Superdex 75 column (GE Healthcare) with a running buffer of 25 mM Tris (pH 8.0), 150 mM NaCl, and 2 mM

dithiothreitol (DTT). The pure fractions were then collected and concentrated for crystallization. Proteins SsBcmC, SoBcmC, SsBcmG, PaBcmG and mutant SsBcmE$^{T307A}$ were overexpressed and purified following the same procedure described above.

### Protein crystallization

The initial crystal screening ultilized commercial kits (QIAGEN, Hampton Research, Rigaku, and Molecular Dimensions), while all optimization reagents were obtained from Sigma-Aldrich. Crystals were obtained using the sitting drop vapor diffusion method at 20 °C. In general, 1 µL protein sample (10 mg mL$^{-1}$) was mixed with 1 µL reservoir solution and equilibrated against 50 µL reservoir solution. The stored protein was diluted by buffer 25 mM Tris (pH 8.0), 150 mM NaCl and 2 mM DTT. The substrate free crystals was gained by incubated with 5 mM αKG and the complex crystals was incubated with 5 mM αKG and 1 mM corresponding substrate on ice for 30 min. The binary complex SsBcmE (with Fe$^{II}$) crystals were grown in 2.5 M Sodium chloride and 0.1 M imidazole hydrochloric acid pH 8.0, and the crystals appeared after almost two months at 20 °C. After trying a lot but failed to determine the complex of SsBcmE, we crystallized several of its mutants and homologous BcmE, and gained mutant SsBcmE$^{T307A}$ complex crystals in anaerobic box after almost two weeks, the condition was 1.26 M Sodium phosphate monobasic monohydrate, 0.14 M Potassium phosphate dibasic, pH 5.6. Crystallization of the SsBcmE$^{T307A}$ complex involved the following steps: First, the aerobically purified protein was placed in an anaerobic chamber. Then, its buffer was replaced with an oxygen-free solution of the same composition. Additionally, the cosubstrate αKG and substrate **1** were prepared by dissolving them in water and DMSO, respectively, ensuring that the oxygen was removed before use. The screening process was the same as above mentioned. The ternary SsBcmC (with Fe$^{II}$ and αKG) was grown in 0.1 M MES/imidazole pH 6.5, 0.03 M of each divalent cation (0.3 M magnesium chloride, 0.3 M calcium chloride) and 10% PEG 20000, 20% PEG MME 550. We could not get any high quality SsBcmC complex crystals, so we tried the homolog proteins from *S. ossamyceticus* and *P. aeruginosa*. The crystals of SoBcmC•Fe$^{II}$•αKG•**2** were grown in 0.2 M Ammonium chloride, 0.1 M Na acetate pH 5, 20% PEG 6000. The crystals of BcmG were also gained by homologous protein from *P. aeruginosa*. The crystals of ternary PaBcmG (with Fe(II) and αKG) were grown in 10% w/v PEG 4000, 20% v/v glycerol 0.03 M of each NPS (0.3 M sodium nitrate, 0.3 M disodium hydrogen phosphate, 0.3 M ammonium sulfate), 0.1 M MES/imidazole pH 6.5. The crystals of PaBcmG•Fe$^{II}$•αKG•**3** were grown in 0.1 M MES pH 6.5, 1.6 M magnesium sulfate. All crystals were flash-frozen in liquid nitrogen using the reservoir solution containing 20% (v/v) glycerol as cryo-protectant.

### Data collection and structure determination

All X-ray diffraction data were collected at the Shanghai Synchrotron Radiation Facility (SSRF). For SsBcmE•Fe$^{II}$ data were collected at the wavelength of 0.9792 Å in beamline BL17U1. Data of SsBcmE$^{T307A}$•Fe$^{II}$•αKG•**1**, SoBcmC•Fe$^{II}$•αKG•**2**, SsBcmC•Fe$^{II}$•αKG and PaBcmG•Fe$^{II}$•αKG were collected at the wavelength of 0.9785 Å in beamline BL19U1. Data of PaBcmG•Fe$^{II}$•αKG•**3** were collected at the wavelength of 0.9792 Å in beamline BL18U1. Data reduction and integration were achieved with HKL3000[86] and the XDS[87] software package. The statistics for data collection are listed in Supplementary Table 1. The phase of SsBcmE•Fe$^{II}$ was solved by molecular replacement in the Phenix-Phaser program[88] using 7V3O as initial searching model. The complex structures SsBcmE$^{T307A}$•Fe$^{II}$•αKG•**1**, SsBcmC•Fe$^{II}$•αKG and PaBcmG•Fe$^{II}$•αKG were determined by molecular replacement using Phaser with SsBcmE•Fe$^{II}$ structure as the initial search model. The quaternary complex structure of SoBcmC and PaBcmG were determined by molecular replacement using Phaser with corresponding substrate free crystal structure as the initial search model. Iterative cycles of model building and refinement were performed in Coot[89] and Phenix[90], respectively. Refinement statistics for each final model are recorded in

Supplementary Table 1. Structure Figures were drawn using PyMol 2.4.1 (Schrodinger, LLC)[91]. For the crystal structure of apo SsBcmE was not intact with some loops and fragments were missing, we built its structure by AlphaFold2 (AF2) (https://colab.research.google.com/github/sokrypton/ColabFold/blob/main/AlphaFold2.ipynb) and found it was very similar to both apo and complex SsBcmE, with r.m.s.d. value 0.57 Å and 0.82 Å, respectively (Supplementary Figs. 6e–g). So we using the AF2 model as the apo structure for the next docking or comparison work. The complex model of SsBcmE$^{F273A}$•Fe$^{II}$•αKG•**1** and SsBcmE•Fe$^{II}$•αKG•**1** was built by Autodock vina[92]. The volume of active pockets was calculated by online sever (http://altair.sci.hokudai.ac.jp/g6/service/pocasa/).

### Quantum chemical computations

Quantum chemical calculations were performed using Gaussian 16 package[93]. Geometry optimizations were conducted at the function of B3LYP-D3[94], and Fe was described via the effective core potential (ECP) of the LanL2DZ[95] and all the other atoms with the 6-31 G(d,p) basis set. Vibrational frequency calculations were carried out at the same level to ensure that all of the stationary points were transition states (one imaginary frequency) or minima (no imaginary frequency) and to evaluate zero-point vibrational energies (ZPVE) and thermal corrections at 298 K. Intrinsic reaction coordinates (IRC) calculations were conducted for all transition states to confirm their connectivity between corresponding minima. Single-point energy calculations were carried out at the B3LYP-D3 level with the SDD[96] for Fe, and the 6-311 + G(2 d,p) basis set for all the other atoms. The B3LYP functional within density functional theory (DFT) has been extensively validated for studies of αKGDs[57,60,62,64,97–100]. Notably, high-level ab initio calculations were also performed and compared with the DFT results by Neese[101] and Borowski[22,102] groups to study the dioxygen activation process of αKGDs. Neese et al.[60,101] found that B3LYP results show excellent agreement with the ab initio calculations. And Borowski et al.[102] revealed that CCSD(T) benchmark calculations support the dispersion corrected B3LYP (B3LYP-D3) spin-state energies. The CPCM[103–105] solvation model was used with chlorobenzene as the solvent that approximates protein environment[99,106,107].

### Molecular dynamics simulations

All molecular dynamics (MD) simulations were conducted using the Amber 16 package[108]. The force field parameters for the iron-containing active site were generated using the Metal Center Parameter Builder (MCPB.py)[109] as implemented in Amber 16 package. Bond and angle force constants were calculated using the Seminario method[110], while atomic point charges were derived from electrostatic potentials using the ChgModB method. The MCPB tools have been successfully used for description of mononuclear non-heme iron center[57,64,111,112]. Substrates **1**–**3** were optimized at the B3LYP-D3/6-31 G(d,p) level using Gaussian 16. The partial charges for these ligands were fitted with HF/6-31 G(d) calculations and the restrained electrostatic potential (RESP)[113,114] protocol implemented by the Antechamber module in Amber 16 package. The force field parameters for these three ligands were adapted from the standard general amber force field 2.0 (gaff2)[115] parameters. Meanwhile, the protein was described using the standard Amber14SB force field. Each system was neutralized by the adding of Na$^+$ ions and subsequently solvated in a rectangular TIP3P[116] water box, with a buffer distance of 10 Å maintained on each side. Each system underwent equilibration through a series of minimizations interspersed with short MD simulations. During this process, restraints on the protein backbone heavy atoms were gradually released, with force constant decreasing from 10, 2, 0.1 and finally 0 kcal mol$^{-1}$ Å$^{-2}$. The system was then gradually heated from 0 to 300 K over 50 ps, during which a restraint of 10 kcal mol$^{-1}$ Å$^{-2}$ was maintained on the protein backbone heavy atoms. Subsequently, a 10 ns MD simulation was conducted at constant temperature and

pressure, with the protein backbone heavy atoms restrained by a force constant of 5 kcal mol$^{-1}$ Å$^{-2}$. Finally, a standard 10 ns MD simulation was performed without restraints. The pressure was maintained at 1 atm using isotropic position scaling, while the temperature was kept at 300 K through Berendsen thermostat method. Long-range electrostatic interactions were handled using the Particle Mesh Ewald (PME)[117] method, with a cutoff distance of 12 Å applied to both PME and van der Waals (vdW) interactions. A time step of 2 fs was employed, in conjunction with SHAKE algorithm applied to hydrogen atoms. Periodic boundary condition was also utilized. The stability of each system, including structure, energy, and temperature fluctuations, was thoroughly examined. Additionally, the convergence of the simulations was assessed by monitoring the root-mean-square deviation (r.m.s.d.) of the structures.

### Reporting summary

Further information on research design is available in the Nature Portfolio Reporting Summary linked to this article.

## Data availability

Atomic coordinates of SsBcmE$^{T307A}$•Fe$^{II}$•αKG•**1**, SsBcmC•Fe$^{II}$•αKG, SoBcmC•Fe$^{II}$•αKG•**2**, PaBcmG•Fe$^{II}$•αKG, PaBcmG•Fe$^{II}$•αKG•**3**, have been deposited in the Protein Data Bank (PDB) under accession codes 8XHY, 8XHP, 8XHQ, 8XHT, 8XHX (https://www.rcsb.org/). The PDB 7V3O could be download from PDB Database (https://doi.org/10.2210/pdb7V3O/pdb). The DNA sequences of genes *SsbcmE*, *SsbcmC*, *SsbcmG*, *SobcmC* and *PabcmG* are downloaded from GenBank with accession numbers MG018995, WP_055518629 and HGP0145857 and the codon-optimized DNA sequences are listed in Supplementary Information. Other data supporting the conclusions of this study are presented in the main text, Supplementary Information and from corresponding author(s) upon request. Source data are provided with this paper.

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

## Acknowledgements

This work was supported by grants from Jiangsu Basic Research Center for Synthetic Biology Grant (BK20233003 to J.Z.), the National Key Research and Development Program of China (2022YFC2303100 to G.-L.T. and 2022YFA1503200 to Y.L.), the National Natural Science Foundation of China (22207117 to J.-B.H., 22077062 to Y.L. and 21825804 to S.Y.), the Natural Science Foundation of Jiangsu Province (BK20230018 to Y.L.) and the Shenzhen Science and Technology Program (ZDSYS20210623091810032 to J.Z.). The authors also thank the staff members of beam lines BL17U1 of Shanghai Synchrotron Radiation Facility, BL18U1 (https://cstr.cn/31129.02.NFPS.BL18U1) and BL19U1 (https://cstr.cn/31129.02.NFPS.BL19U1) of National Facility for Protein Science in Shanghai for access and help with the X-ray data collection. We thank the High Performance Computing Center (HPCC) of Nanjing University for doing the numerical calculations in this paper on its blade cluster system.

## Author contributions

S.Y., Y.L., G.-L.T. and J.Z. conceived the project. L.W. performed all crystallographic studies. J.-B.H. performed all in vitro biochemical experiments. W.W. conducted all computational studies. H.-X.P. and X.W. performed protein purification. All authors analyzed and discussed the results. L.W., J.-B.H., W.W., S.Y., Y.L., G.-L.T., and J.Z. prepared the manuscript.

## Competing interests

The authors declare no competing interests.
