## [Transparent Peer Review file · Nature Communications]

Three distinct strategies lead to programmable aliphatic C-H oxidation in bicyclomycin biosynthesis

Corresponding Author: Professor Jiahai Zhou

Version 0:

Reviewer comments:

Reviewer #1

(Remarks to the Author)

In this manuscript, the authors report the mechanistic analysis of three iron/ α -ketoglutarate-dependent dioxygenases involved in the biosynthesis of bicyclomycin and the engineering of these enzymes. The enzymes, BcmE, BcmC, and BcmG, selectively hydroxylate specific C-H bonds on similar substrates. Structural, computational, and mutagenesis studies revealed distinct strategies employed by each enzyme for regioselectivity. The mutagenesis experiments rationally altered the regioselectivity of the hydroxylation reaction. Furthermore, the authors also demonstrated the site-selective and stereoselective hydroxylation of various isoleucyl-containing cyclodipeptides. The manuscript is well written and illustrated. This work provides a detailed molecular understanding of enzymatic selectivity, contributing to synthetic chemistry and expanding the application of biocatalysts. The results presented in this study are likely to capture the interest of researchers in the fields of biosynthesis, biocatalysis and natural product chemistry. Therefore, I recommend the publication of this paper after they revised some points to improve the manuscript.

1. Are there any desaturated products generated by mutant enzymes? The authors should also discuss how the enzyme controls the oxygen rebound reaction and desaturation reaction.
2. QM/MM calculations or DFT calculations including active site residues should be performed and compared to the minimal theoretical DFT results.
3. Figure 3: It should be clearly noted what the density maps of ligands are. Are these Fo-Fc omit maps?
4. Figure 4: The right panels are not informative. Generally, the iron binding site should be inside the enzyme, but these figures seem to represent the iron center on the surface of the enzyme.
5. The mechanism for the formation of 3b/c should be discussed.
6. It should be explained how the authors achieved the complex structures with the SsBcmE T307A mutant.

Reviewer #2

(Remarks to the Author)

General summary: This manuscript by Wu et al. presents evidence for how three Fe- and α KG-dependent hydroxylases, BcmE, BcmC, and BcmG, achieve distinct regioselectivity in C-H bond hydroxylation in bicyclomycin biosynthesis. The similarity in substrate scope for these enzymes offers a unique opportunity for dissecting enzymatic control of C-H activation. The authors identify two main concepts to rationalize activities. BcmC targets a different type of C-H bond (tertiary versus primary for BcmE/BcmG). Consequently, the innate reactivity of the target C-H bonds explains why this enzyme targets its preferred site. The authors calculated free energy barriers for C-H cleavage by the ferryl intermediate in this enzyme, showing that tertiary C-H bonds are indeed the most susceptible to hydrogen atom transfer when only the primary coordination sphere is considered. To further examine the control of C-H cleavage selectivity in the other enzymes, the authors used x-ray crystal structures of reactant complexes, with Fe(II), α KD, and/or their prime substrates bound. While sharing very similar overall folds, the prime substrates are poised differently in the active site with respect to the iron centers, allowing distinct C-H bond to be cleaved and hydroxylated in each case. With these static structures as starting points, the authors employed MD simulations to assess the feasibility of C-H cleavage for each bond based on geometric constraints. Site-directed mutagenesis was used to evaluate the robustness of the regioselectivity when each interaction with the prime substrate was modulated. While BcmC followed the innate reactivity of C-H bonds and was indifferent to mutations, the

selectivity of BcmE and BcmG was compromised by replacements hydrophobic contacts and a particular tyrosine that H-bonds to a key -OH group in the substrate. These results led the authors to conclude that BcmE, BcmC, and BcmG leveraged steric hinderance, innate reactivity, and a directing group (hydrogen bond), respectively, to enforce regioselectivity. The proposed strategies were further confirmed by testing reaction outcomes with non-native diketopiperazines. The experimental work is comprehensive – but the study builds on long-standing ideas proposed by others about the importance of substrate positioning and inherent reactivity. Thus the novelty is not high and the conclusions are somewhat overblown without sufficient credit given to prior work in this area through literature citations. Another major weakness of the work is that, despite the impressive array of experiments carried out by the authors, the logical flow of the arguments and the support by the experimental data are not fully clear.

Major issues to be resolved are listed below:

- (1) In the introduction, the discussion of control of reaction outcome in metalloenzymes that target the same substrate with different regiochemistry is not sufficiently comprehensive. Additional examples exist – see VioC and OrfP for Fe/2OG enzymes, for example. Furthermore, study of Fe/2OG enzymes that operate on the same or similar substrates with different outcomes has already established the importance of substrate positioning. And other studies have also addressed the contribution of innate reactivity in controlling outcome. These should all be acknowledged and appropriately referenced in the introduction.
- (2) In Fig. 1b 3), please adopt the consistent numbering for carbon atoms as used throughout the manuscript.
- (3) The introduction to constructs used in crystallography is confusing (starting on line 111 on page 7). The authors begin with an introduction to the Ss orthologs but mixed in discussion with rationales behind the final choices that include enzymes from other organisms and mutations. It would be more helpful to start with SsBcmE T307A, SsBcmC, SoBcmC, and PaBcmG (as listed in Extended Data Table 1) and then explain the rationale behind each choice. SsBcmG and wild-type SsBcmE did not crystallize, and the quaternary complex was unattainable with SsBcmC. The perturbed selectivity for SsBcmE T307A (Extended Data Fig. 2) should also be mentioned early. It is not clear how representative its structure would be of its wild-type counterpart. That being said, the ability of this team to overcome challenges in crystallographic characterization is commendable – but the logic needs to be more clearly explained and the disadvantages of using variants needs to be acknowledged.
- (4) On the same note, Supplementary Fig. 6 should include the sequence of SsBcmC, as its tertiary complex structure is solved. In line 118, the words “apo” were used but in a confusing manner; none of the solved structures were free of iron and the co-substrate α KG. For the zoom-in views in Fig. 3, the magenta and blue colors have rendered oxygen and nitrogen atoms hard to see, respectively. It could also be useful to adjust the clipping planes in PyMOL so that the relative depths of residues can be perceived by a reader.
- (5) In Figure 3, important information is missing that is essential for understanding the key points of the study. Most importantly, an electron density map is shown for the substrate – but the figure caption makes no mention of the type of map or the contour level. Ideally, omit maps should be shown at a minimum of 3.0 sigma. And it would also be optimal show omit maps for the 2OG cosubstrate and any water ligands in the SI. In Fig. 3, I would also recommend showing the metal coordination sphere in the standard orientation with the distal His (higher number) ligand in the lower axial position and the substrate binding site located in the upper axial position. It would also be ideal to show distances between the iron and the nearest carbon atoms in the substrate – inclusion of this information would also help address some of the issues mentioned below.
- (6) In the BcmE T307A crystal structure, C-5a was found to be closest to the iron center among all carbon atoms (line 146, page 9). However, according to Extended Data Fig. 2, BcmE T307A still largely prefers the native selectivity (at C7; product 2) than C-5a (product 1a), assuming the area ratio under the HPLC traces reflects the molar ratio of molecules. This was attributed to the enlarged pocket that allowed for C4-C5 rotation in the T307A variant (line 148) and could be accounted for with MD simulations. While it is understandable that the dominant conformations do not reflect the actual active conformations (i.e., states that are ready to undergo hydrogen atom transfer), in Extended Data Fig. 4e and f, the active conformations in BcmE T307A seemed more prevalent for C5a than C7, seemingly contradictory to the observed selectivity. Are there any explanations to reconcile these data and computational results? Furthermore, does MD address that why the particular C4-C5 rotamer is preferred in crystallo, bringing C-5a closest to the active site?
- (7) Likewise, C-3' was found to be almost equally but slightly closer to the active site than C-2' in the SoBcmC crystal structure (line 165), while C-2' hydroxylation was observed to be the sole outcome (Extended Data Fig. 5). This “contradiction” was again attributed to rotation about the C1'-C2' bond (line 167). If BcmC were to follow the innate reactivity of C-H bonds in the substrate as the authors proposed (Fig. 2b), is substrate positioning still a critical argument (Extended Data Fig. 3b and e) to understand the selectivity of BcmC? If so, how is C-3' accounted for in Extended Data Fig. 3e, and how does MD simulation account for the substrate rotamer observed in the crystal structure? Also, it is desirable to show the activity and selectivity of SsBcmC in Extended Data Fig. 5.
- (8) For PaBcmG, additional minor products (4a/4b) were found for wild-type enzyme in Extended Data Fig.6 but not for other variants. Do those minor products exist for SsBcmG, or are they exclusive to PaBcmG? What are the identities of these minor products, as I cannot find any clue throughout the manuscript?
- (9) BcmC selectivity was largely unperturbed by amino-acid replacements around the active site (Extended Data Fig. 5). In contrast, substrate positioning was critical to BcmE and BcmG, as their selectivity was sensitive to replacements of hydrophobic residues (Extended Data Fig. 2) and Y288 (Extended Data Fig. 6), respectively. The authors claimed that these observations can be attributed to three mutually orthogonal strategies separately adopted by the three enzymes (line 323, page 17), but I find that hard to be concluded solely based on the mutagenesis results, especially the sensitivity towards Y308 in BcmE (Extended Data Fig. 2). The authors have also written that “a cumulative set of interactions” (line 318) led to enzymatic selectivity. Is it warranted to only include steric hinderance control as opposed to directing group control for BcmE? On the other hand, I do believe that BcmG invokes directing group control, but it only becomes convincing when the impressive data from non-native substrates are shown (Fig. 5c). I find it speculative (as the authors have also written in line 268) before the strongest evidence is presented, so is it possible to write a section as a last paragraph of the introduction to briefly walk the readers through the logical flow of the manuscript? In addition, in line 290, the neighboring position should

be “vicinal” rather than “ortho”, as the functional groups are not part of an aromatic system.

(10) Finally, the section titles can be more explicit, especially “BcmE and BcmG employ different strategies to control site selectivity compared to BcmC” (line 87). I suggest “BcmE and BcmG defy the innate reactivity preferences of BcmC”, or it becomes too similar to “BcmE, BcmC and BcmG use three different strategies to control the site selectivity of C-H oxidation” (line 189).

Minor comments:

(1) Page 4, line 60: “ α KGDs amino acid hydroxylases”. Since α KGDs are already defined as “non-heme Fe(II)/ α -ketoglutarate-dependent dioxygenase” (page 3, line 54), consider “ α KG-dependent amino acid hydroxylases”.

(2) Page 4, line 65: “homologous enzymes that catalyzed C-H hydroxylation”. Should be “catalyze”.

(3) Fig. 1b1, b2, b3 figure titles: “homolog enzyme”. Should be “homologous enzymes”. Similarly for the figure captions.

(4) Page 5, line 78: “In our previously biosynthetic studies”. Should be “previous”.

(5) Page 5, starting line 90: The incorporation of a truncated theozyme is an interesting way to quantify the intrinsic reactivity, but it took me a while to realize its importance. Can the readers be more explicit on: i. theozyme only considers primary coordination sphere such that the substrate could possibly be accessed from all orientations by the iron center, ii. inherent site selectivity is based on bond dissociation energy (cite Supplementary Fig. 1).

(6) Fig. 2a: How is this ferryl orientation (i.e., trans to this particular histidine) chosen? What is the evidence of a water ligand?

(7) Fig. 2b title: “different types of hydroxylation (kcal/mol)”. Should be “hydrogen abstraction”, unless the subsequent oxygen rebound step is considered.

(8) Fig. 2b caption title: “Gibbs free energies ... for the hydroxylation reactions”. Consider “Gibbs free energy barriers ... for the hydrogen abstraction reactions”. Also, there needs to be an ending parenthesis after “iron: cyan”. “The energy barriers of every C-H bond were” mixes plural and singular forms.

(9) Page 7, line 119: what is the purpose of italicizing “X” in “HXD-Xn-H”?

(10) Page 7, line 122: “r.m.s.d.” should be “r.m.s.d.”, and there are too many significant digits as the resolution should not allow for accuracy to the third decimal place. Consider two decimal places throughout the manuscript.

(11) Fig.3 caption title: “Fe(II) and α KG dependent”. Consider “Fe(II)- and α KG-dependent”. In addition, “magentas”, “cyans”, and “oranges” should be singular and please correct the spelling of magenta.

(12) Page 8, line 140: it seems odd to characterize the interaction between carbonyl and F273 as π - π stacking.

(13) Page 8, line 142: no need to write “respectively” in this sentence.

(14) Page 9, line 150: “wildtype” should be “wild-type”.

(15) Page 9, line 151: “T307 could pull C7 directing to the iron center”. Consider “could bring C7 closer to the iron center”.

(16) Page 9, lines 152-153: “Residues M177, ... portion of 1.” Could cite Fig. 3a here.

(17) Page 9, line 157: “we infer ...”. This language seems a bit too strong without the upcoming mutagenesis studies. Maybe “hypothesize” is a better word.

(18) Page 10, line 178: is S175 not shown in Fig. 3c?

(19) Page 10, line 179: no need to write “respectively” in this case.

(20) Page 11, line 197: “three enzymes”. Consider “these enzymes” or “these three enzymes”.

(21) Page 14, line 244: “easily reactive”. Consider “inherently more reactive”.

(22) Lines 470 and 484, page 23: “was/were stored in our laboratory”. Consider “was/were previously reported by our laboratory”.

(23) Extended Data Figs. 2, 5, and 6: it might be useful to number the modified carbons for the molecular structures.

(24) Extended Data Table 1: “C121” and “P121” should be “C21” and “P21”, respectively.

(25) Supplementary Fig. 1: “viarous” should be “various”.

(26) Supplementary Fig. 5d: “C-termonal” should be “C-termini”.

(27) Supplementary Fig. 5b figure caption: the use of “2OG” is not consistent with the text (“ α KG”).

(28) Supplementary Fig. 5c figure caption: “palecyan” should be “pale cyan”.

(29) Supplementary Fig. 5d figure caption: “the coarser the higher”. Consider “the thicker the higher”.

(30) Supplementary Fig. 7 figure caption: “magentas” should be “magenta” and “palecyan” should be “pale cyan”.

Reviewer #3

(Remarks to the Author)

In this work a crystallography and computational study is reported on understanding of three nonheme iron dioxygenases that react with the same substrate but give different product distributions. They trap three crystals of structures of the three isozymes and do MD simulations to understand the reactivity differences. Also substrate analogues are tested that give further insight into the C-H activation pathways. Overall it is interesting work that fits the remit of Nature Communications.

1. It is generally accepted that the second-coordination sphere in nonheme iron dioxygenases determines substrate binding and selectivity. This will be useful to mention here and give appropriate citations. I find it difficult to see from Figure 3 what the difference in second coordination sphere is between the various structures. What determines the substrate binding and positioning?

2. The authors start with a set of minimal cluster DFT studies on the first coordination sphere only. It has been shown by several groups that these small cluster models often give the wrong energetics and selectivity (the authors may want to mention this), and indeed the energies in Figure 2 show that the DFT does not predict the selectivity correctly. It would have been better if large DFT clusters had been used based on the new crystal structures. Do these larger structures give insight into the selectivity of the enzymes?

3. Is the polarity different in the active site pockets of the three enzymes?
4. How do the structure of the pdb's compare with other pdb's in the protein databank?
5. How do the DFT calculations compare with DFT and QM/MM studies from the literature? There is a lot of work in the literature by Christov, de Visser, Borowski, Gauld and others.
6. The MD simulations seem to imply that substrate is at quite a large distance from the oxidant, which may be a problem for catalysis. This may require some discussion.

Reviewer #4

(Remarks to the Author)

Version 1:

Reviewer comments:

Reviewer #1

(Remarks to the Author)

The authors have satisfactorily addressed this reviewer's comments, and the manuscript is recommended for acceptance with one suggestion: Figure R2 should be included in the manuscript as a supplementary figure.

Reviewer #2

(Remarks to the Author)

The authors have submitted a revision that addresses prior critiques. The authors have devoted an introductory paragraph for an overview of the logical framework in this manuscript – although this paragraph could include more details to help the reader understand exactly which experiments or results were key in defining the proposed reaction mechanisms.

A few remaining questions include the major and minor issues outlined below.

In the description of the theozyme (line 109), why is a water ligand included in the model of the reactive intermediate? The last sentence (lines 115-118) in this section is not comprehensible. I think a summary statement is needed to sum up the findings from the computational assessment. It seems that for all enzymes evaluated, the secondary carbon targets yield lower energy barriers – and this is sensible because target of this position by HAT would result in a more stable radical. Then state that BcmC targets the most reactive position but BcmE and BcmG react with sites that are intrinsically less reactive. But I think it should be mentioned here that the comparisons made (C2' versus C7 and C5 versus C3') are on completely different sides of the molecule. And it is common knowledge that Fe/2OG, P450, and many other enzymes that employ reactive iron intermediates use substrate positioning to target specific sites that may not be the most reactive on a given substrate. I think the relevant literature precedent should be cited here because this is not a new idea.

The sentences in lines 195-199 do not make sense. The first sentence states that binding substrate yields “a more compact active site” but the overlay of “apo” and substrate bound structures in figure S7d does not show any significant conformational changes in the overall fold. It does appear that the C-terminus of the protein might order to cover up the active site- but I would not call that a compaction of the active site – it is simply covering it up or shielding from solvent. Additionally, the authors speculate that this conformational change that occurs on binding of substrate somehow promotes dissociation of a water ligand and binding of O₂ to the iron cofactor. However, the complex with substrate still shows a water ligand bound. I don't think this is worthy of comment. Additionally, the appropriate references for this mechanism of substrate triggered O₂ addition are not included. Some review articles are cited but the original research articles that give rise to this mechanism previously proposed by others are not referenced.

In the section starting on page 12, it would be helpful to include information about yield when relevant. For example, the reactions of BcmE with variants that dramatically alter regioselectivity (giving 1b/c products) also show lesser amounts of product accumulation. It would also be useful to include more structural context for these results. Why do the authors think that the other side of the molecule can be targeted when certain mutations are made to open up the active site? In general this section would benefit from adding more context about how the mutagenesis and MD results fit in with the previously discussed structural work.

One minor but serious issue that still remains in spite of the provided response (see Reviewer 2, minor comments, bullet points 7 and 8) is that “hydroxylation” should be replaced with “hydrogen abstraction” in both the header of Fig. 2b and the figure caption of Fig. 2. Regarding bullet point 10 in the minor comments, not all r.m.s.d. values have been truncated to two decimal places (e.g., caption of Supplementary Fig. 7). The newly added sentence (lines 62-65): “... such like ... Proline” has serious grammatical issues.

Reviewer #3

(Remarks to the Author)

All issues have been addressed well, publication is recommended.

Version 2:

Reviewer comments:

Reviewer #2

(Remarks to the Author)

The authors have generally addressed the previous concerns.

On lines 115-118 of the previous submission, the authors have added some references to research articles describing the importance of substrate positioning in determining reaction outcome in Fe/2OG enzymes. Please add Matthews et al. (2009) "Substrate positioning controls the partition between halogenation and hydroxylation in the aliphatic halogenase SyrB2." PNAS 106, 17723.

Responses to Reviewers' comments

Remarks from the reviewers are shown in black. Our responses are shown in blue.

Reviewer #1 (Remarks to the Author):

In this manuscript, the authors report the mechanistic analysis of three iron/ α -ketoglutarate-dependent dioxygenases involved in the biosynthesis of bicyclomycin and the engineering of these enzymes. The enzymes, BcmE, BcmC, and BcmG, selectively hydroxylate specific C-H bonds on similar substrates. Structural, computational, and mutagenesis studies revealed distinct strategies employed by each enzyme for regioselectivity. The mutagenesis experiments rationally altered the regioselectivity of the hydroxylation reaction. Furthermore, the authors also demonstrated the siteselective and stereoselective hydroxylation of various isoleucyl-containing cyclodipeptides. The manuscript is well written and illustrated. This work provides a detailed molecular understanding of enzymatic selectivity, contributing to synthetic chemistry and expanding the application of biocatalysts. The results presented in this study are likely to capture the interest of researchers in the fields of biosynthesis, biocatalysis and natural product chemistry. Therefore, I recommend the publication of this paper after they revised some points to improve the manuscript.

1. Are there any desaturated products generated by mutant enzymes? The authors should also discuss how the enzyme controls the oxygen rebound reaction and desaturation reaction.

R: We did not detect any desaturated products by mutant enzymes. Inspired by the mechanisms of hydroxylation and desaturation in reported works of literature (*Crit. Rev. Biochem. Mol. Biol.* **39**, 21–68 (2004); *Eur. J. Inorg. Chem.* **21**, 4245–4254 (2005); *Acc. Chem. Res.* **40**, 484–492 (2007); *Chem. Sci.* **15**, 3466–3484 (2024); *J Am Chem Soc.* **140**, 7116–7126 (2018).; *Science.* **343**, 1140–1144 (2014).; *Biochemistry* **62**, 229–240

(2023)), we summarized the mechanism of oxygen rebound reaction leading to hydroxylation products as follows: 1) α KG binds to the Fe(II) center in a bidentate configuration, with its keto group opposite the Asp/Glu and its carboxylate opposite one of the His residues, thus displacing two metal-bound water molecules (**Fig. R1a**). 2) Upon binding of the primary substrate (RH) to the enzyme active site (not, however, to the metal ion), the third metal-bound water is displaced (**Fig. R1a**). Substrate binding triggers the process of creating a site for binding an O₂ molecule, generating a Fe(III)-superoxo intermediate (**Fig. R1a**). 3) The distal oxygen atom of the Fe(III)-superoxo species attacks C2 of α KG to yield a peroxohemiketal bicyclic intermediate (**Fig. R1a**). 4) This species initiates the oxidative decarboxylation of α KG to release CO₂ and yield, perhaps via an Fe(II)-peracid intermediate (not shown), an Fe(IV)-oxo species (known as the ferryl intermediate) with bound succinate (**Fig. R1a**). 5) The ferryl species abstracts a hydrogen atom from the primary substrate to generate Fe(III)-OH and a substrate radical (**Fig. R1a**). The process of desaturation was branched after step 5. In the hydroxylation pathway, the carbon radical attacks the oxygen ligand to form a new C-O bond. This radical-coupling step, commonly referred to as oxygen rebound, regenerates the Fe(II) state of the cofactor for subsequent turnover. In the desaturation process, there is a second HAT from the adjacent carbon to the Fe(III)-OH complex. Then there are three ways to form a C=C bond, converting directly to double dehydrogenation product (**Fig. R1b** pathway A). Two other most likely pathways for conversion of the radical to product would involve rebound to the carbon followed by dehydration and adjacent deprotonation (**Fig. R1b** pathway B) or electron transfer from the radical to the Fe(III)-OH cofactor followed by the adjacent carbon deprotonation (**Fig. R1b** pathway C)¹.

As the purpose of this manuscript is not the mechanism of hydroxylation, we add several citations as follow.

43. Islam, M. S., Leissing, T. M., Chowdhury, R., Hopkinson, R. J. & Schofield, C. J. 2-Oxoglutarate-Dependent Oxygenases. *Annu. Rev. Biochem.* **87**, 585-620 (2018).

48. Ali, H. S., Henchman, R. H., Warwicker, J. & de Visser, S. P. How Do

Electrostatic Perturbations of the Protein Affect the Bifurcation Pathways of Substrate Hydroxylation versus Desaturation in the Nonheme Iron-Dependent Viomycin Biosynthesis Enzyme? *The Journal of Physical Chemistry A* **125**, 1720-1737 (2021).

55. Saridakis, E. et al. Cryo-EM structure of transcription termination factor Rho from *Mycobacterium tuberculosis* reveals bicyclomycin resistance mechanism. *Commun. Biol.* **5**, 120 (2022).

56. Carrano, L. et al. Effects of bicyclomycin on RNA- and ATP-binding activities of transcription termination factor Rho. *Antimicrobial Agents and Chemotherapy* **42**, 571-578 (1998).

57. Meng, S. et al. A Six-Oxidase Cascade for Tandem C-H Bond Activation Revealed by Reconstitution of Bicyclomycin Biosynthesis. *Angew Chem. Int. Ed.* **57**, 719-723 (2018).

[figure redacted]

Fig. R1 The mechanism of non-heme FeII/ α -ketoglutarate-dependent dioxygenase catalyzed (a) hydroxylation and (b) destaturation¹.

2. QM/MM calculations or DFT calculations including active site residues should be performed and compared to the minimal theoretical DFT results.

R: We thank the reviewer for the suggestion. QM/MM calculations or DFT calculations including the active site residues are primarily intended to explain the underlying principles and controlling factors of the protein scaffold for reactivity and selectivity but are different from our aim of resolving the intrinsic site selectivity of the substrate.

To reveal the inherent site selectivity of the C(sp^3)-H hydroxylation in the cognate substrates (**1-3**), a truncated catalytic-residue theozyme model was constructed based on the reported spectra data, crystal structure, and computational simulations (*Nat. Commun.* **9**, 1168 (2018); *J. Am. Chem. Soc.* **140**, 7116-7126 (2018); *Phys. Chem. Chem. Phys.* **19**, 20188-20197 (2017).; *Nat. Catal.* **6**, 637-648 (2023)). In detail, this model contained a cognate substrate, Fe^{IV}-oxo species, two methylimidazoles for two histidine

residues, two acetate anions in aspartic acid and succinate, as well as one water. The theozyme model only considered the primary coordination sphere such that the substrate could be accessed from all orientations by the iron center, to quantitatively evaluate the inherent reactivity of the C(sp^3)-H bond by using the hydrogen transfer energy barrier of the substrate. As pointed out by **reviewer #2**, “the incorporation of a truncated theozyme is an interesting way to quantify the intrinsic reactivity. This theozyme model only considers the primary coordination sphere such that the substrate could be accessed from all orientations by the iron center. And the inherent site selectivity is based on bond dissociation energy (**Supplementary Fig. 1**).”

Supplementary Fig. 1. The bond dissociation energy of different types of C-H.

Similarly, DFT calculations based on theozyme models have been used to reveal the inherent size selectivity in ring-closure reactions as well as regioselectivity and periselectivity among pericyclic reactions, etc., as detailed in our previous work and in K.N. Houk's articles (*Nat. Catal.* **6**, 637-648 (2023); *Nature* **586**, 64-69 (2020); *Nat. Catal.* **4**, 1059-1069 (2021).; *Nat. Catal.* **4**, 223-232 (2021)).

3. Figure 3: It should be clearly noted what the density maps of ligands are. Are these Fo-Fc omit maps?

R: We thank the reviewer for the comment. The density maps were Fo-Fc omit maps,. we have added a depiction of the density in the revised manuscript as follows “**The Fo – Fc (omit map) electron density map (blue mesh) for substrates in active site is**

contoured to 3.0σ ." For better understanding, we also re-drew Fig. 3, especially the zoom-in pictures.

Revised Fig. 3. The substrate-bound complex structures reveal detailed active site of Fe(II) and α KG dependent SsBcmE, SoBcmC and PaBcmG. a, b, c, The active pocket of the complex structures of SsBcmE^{T307A}•Fe^{II}• α KG•1 (green), SoBcmC•Fe^{II}• α KG•2 (pink) and PaBcmG•Fe^{II}• α KG•3 (blue), respectively. The above was the overall structure and the below was the detailed interactions of corresponding substrates and residues, shown as sticks. The Fo-Fc (omit map) electron density map (blue mesh) for substrates in the active site is contoured to 3.0σ . The oxygen atoms were colored in red; nitrogen atoms were colored in blue and the carbon atoms of α KG, 1, 2, and 3 were colored in yellow, magenta, cyan, and orange.

4. Figure 4: The right panels are not informative. Generally, the iron binding site should be inside the enzyme, but these figures seem to represent the iron center on the surface of the enzyme.

R: The right panels display a schematic representation focusing on the key residues within the active pockets that influence selectivity, not showing the whole enzyme molecules. For better understanding of the figures, we have added a description to the

figure legend of Fig. 4 in the revised manuscript as follows: “**In the right panels, we present simplified models of the active pockets to clarify the selectivity mechanisms. It's important to note that these models do not represent the full enzymes, and the iron centers are embedded within the active pockets, not exposed on the enzyme surface.**”

5. The mechanism for the formation of 3b/c should be discussed.

R: We thank the reviewer for the critical suggestion. **3b** and **3c** are two overoxidation diastereomer products catalyzed by PaBcmG variant (Y288F). According to our previous research as well as other group studies on BcmG-catalyzed reaction¹⁻³, we proposed that PaBcmG-Y288F can catalyze the further oxidation of product **3a** at C3' to produce an aldehyde-containing intermediate, which undergoes addition by amide N2 to form **3b** and **3c** (Fig. R2). In the revised manuscript, we have added the relevant description for the formation of products **3b** and **3c** to the legend of **Extended Data Fig. 6** as follows: “**Similar to 4a and 4b produced by WT PaBcmG and SsBcmG, the diastereomers 3b and 3c are also two overoxidation products. According to our previous research as well as other group studies on BcmG-catalyzed reaction²⁻⁴, we proposed that PaBcmG-Y288F can catalyze the further oxidation of product 3a at C3' to produce an aldehyde-containing intermediate, which undergoes addition by amide N2 to form 3b and 3c.**”.

[figure redacted]

Fig. R2. Proposed formation mechanism of **3b/c** produced by BcmG Y288F mutant.

6. It should be explained how the authors achieved the complex structures with the SsBcmE T307A mutant.

R: We thank the reviewer for the comment and suggestion. We have added more details of achieving the complex structures with the SsBcmE^{T307A} mutant in the method part as follows: “**Crystallization of the SsBcmE^{T307A} complex involved the following**

steps: First, the aerobically purified protein was placed in an anaerobic chamber. Then, its buffer was replaced with an oxygen-free solution of the same composition. Additionally, the cosubstrate α KG and substrate 1 were prepared by dissolving them in water and DMSO, respectively, ensuring that the oxygen was removed before use. The screening process was the same as above mentioned.”

Reviewer #2 (Remarks to the Author):

This manuscript by Wu et al. presents evidence for how three Fe- and α KG-dependent hydroxylases, BcmE, BcmC, and BcmG, achieve distinct regioselectivity in C-H bond hydroxylation in biocyclomycin biosynthesis. The similarity in substrate scope for these enzymes offers a unique opportunity for dissecting enzymatic control of C-H activation. The authors identify two main concepts to rationalize activities. BcmC targets a different type of C-H bond (tertiary versus primary for BcmE/BcmG). Consequently, the innate reactivity of the target C-H bonds explains why this enzyme targets its preferred site. The authors calculated free energy barriers for C-H cleavage by the ferryl intermediate in this enzyme, showing that tertiary C-H bonds are indeed the most susceptible to hydrogen atom transfer when only the primary coordination sphere is considered. To further examine the control of C-H cleavage selectivity in the other enzymes, the authors used x-ray crystal structures of reactant complexes, with Fe(II), α KD, and/or their prime substrates bound. While sharing very similar overall folds, the prime substrates are poised differently in the active site with respect to the iron centers, allowing distinct C-H bond to be cleaved and hydroxylated in each case. With these static structures as starting points, the authors employed MD simulations to assess the feasibility of C-H cleavage for each bond based on geometric constraints. Site-directed mutagenesis was used to evaluate the robustness of the regioselectivity when each interaction with the prime substrate was modulated. While BcmC followed the innate reactivity of C-H bonds and was indifferent to mutations, the selectivity of BcmE and BcmG was compromised by replacements hydrophobic contacts and a particular

tyrosine that H-bonds to a key -OH group in the substrate. These results led the authors to conclude that BcmE, BcmC, and BcmG leveraged steric hinderance, innate reactivity, and a directing group (hydrogen bond), respectively, to enforce regioselectivity. The proposed strategies were further confirmed by testing reaction outcomes with non-native diketopiperazines. The experimental work is comprehensive – but the study builds on long-standing ideas proposed by others about the importance of substrate positioning and inherent reactivity. Thus the novelty is not high and the conclusions are somewhat overblown without sufficient credit given to prior work in this area through literature citations. Another major weakness of the work is that, despite the impressive array of experiments carried out by the authors, the logical flow of the arguments and the support by the experimental data are not fully clear.

R: We thank the reviewer for the critical comments and valuable suggestions.

Enzymes are evolved by nature and often exhibit excellent regio- and stereoselectivities, rendering them powerful catalysts for the application of C(sp³)-H functionalization. Despite numerous enzymes including Fe^{II}/ α -ketoglutarate-dependent dioxygenase (α KGD) have been identified in various natural product biosynthetic pathways, which perform different C-H functionalization reactions such as hydroxylation with different control strategies⁵⁻¹⁹, the example of homologous enzymes that catalyzed C-H hydroxylation with different site-selectivity in the same biosynthetic pathway is extremely rare. Site- and stereoselective C-H functionalization of unactivated sp³-carbons remains a formidable challenge in the field of synthetic chemistry. In our previous study, we discovered that three α KGDs, BcmE, BcmC and BcmG, catalyze the site- and stereo-selective installation of hydroxyl groups on inert C-H bonds of the cyclodipeptide scaffold involved in bicyclomycin biosynthesis²⁰. However, the detailed mechanisms of catalysis and selectivity controlled by these three enzymes remain unclear. In this work, we focus on elucidating how these three α KGDs employ different strategies to control the site selectivity of C-H hydroxylation reactions. Through crystallographic studies, computational simulations, and biochemical experiments, we revealed that BcmE, BcmC, and BcmG precisely tune site-selective C-H hydroxylation

by using three mutually orthogonal strategies: steric hindrance control, substrate innate control, and directing group control. Although these corresponding strategies have been mentioned in the various enzyme-catalyzed hydroxylation²¹⁻²⁶, to our knowledge, this is the first example of homologous enzymes that catalyzed C–H activation with three different selectivity models in the same biosynthetic pathway. We believe that this work will be of broad general interest in developing new biocatalysts to address the site- and stereoselective functionalization of C–H bonds.

Major issues to be resolved are listed below:

(1) In the introduction, the discussion of control of reaction outcome in metalloenzymes that target the same substrate with different regiochemistry is not sufficiently comprehensive. Additional examples exist – see VioC and OrfP for Fe/2OG enzymes, for example. Furthermore, study of Fe/2OG enzymes that operate on the same or similar substrates with different outcomes has already established the importance of substrate positioning. And other studies have also addressed the contribution of innate reactivity in controlling outcome. These should all be acknowledged and appropriately referenced in the introduction.

R: We are grateful to the reviewer for pointing out the missing references. We have added the relevant references into the introduction part and made supplement to the corresponding description as follows: **“Several α KG-dependent amino acid hydroxylases could hydroxylate the same amino at different sites, such like VioC, OrfP, CmnC and EFEs catalyzing the hydroxylation of L-Arg or analogue, P3H and P4H could selectively oxidize C-H bonds of Proline.**^{41-50”}. The added references were as follow.

46. Ali, H. S.; de Visser, S. P., Catalytic divergencies in the mechanism of L-arginine hydroxylating nonheme iron enzymes. *Front Chem.* **12**, 1365494 (2024).

47. Hsiao, Y. H.; Huang, S. J.; Lin, E. C.; Hsiao, P. Y.; Toh, S. I.; Chen, I. H.; Xu, Z.; Lin, Y. P.; Liu, H. J.; Chang, C. Y., Crystal structure of the α -ketoglutarate-dependent non-heme iron oxygenase CmnC in capreomycin biosynthesis and its engineering to

- catalyze hydroxylation of the substrate enantiomer. *Front. Chem.* **10**, 1001311 (2022).
48. Ali, H. S.; Henschman, R. H.; Warwicker, J.; de Visser, S. P., How Do Electrostatic Perturbations of the Protein Affect the Bifurcation Pathways of Substrate Hydroxylation versus Desaturation in the Nonheme Iron-Dependent Viomycin Biosynthesis Enzyme? *J. Phys. Chem. A* **125** (8), 1720-1737 (2021).
49. Mitchell, A. J.; Dunham, N. P.; Martinie, R. J.; Bergman, J. A.; Pollock, C. J.; Hu, K.; Allen, B. D.; Chang, W. C.; Silakov, A.; Bollinger, J. M., Jr.; Krebs, C.; Boal, A. K., Visualizing the Reaction Cycle in an Iron(II)- and 2-(Oxo)-glutarate-Dependent Hydroxylase. *J. Am. Chem. Soc.* **139** (39), 13830-13836 (2017).
50. Peters, C.; Buller, R. M., Industrial Application of 2-Oxoglutarate-Dependent Oxygenases. **9** (3), 221 (2019).

(2) In Fig. 1b 3), please adopt the consistent numbering for carbon atoms as used throughout the manuscript.

R: We have taken the suggestion and corrected the corresponding carbon number, see revised Fig. 1b 3).

Revised Fig. 1b 3). The strategies to catalyze C-H activation.

(3) The introduction to constructs used in crystallography is confusing (starting on line 111 on page 7). The authors begin with an introduction to the Ss orthologs but mixed in discussion with rationales behind the final choices that include enzymes from other organisms and mutations. It would be more helpful to start with SsBcmE T307A, SsBcmC, SoBcmC, and PaBcmG (as listed in Extended Data Table 1) and then explain

the rationale behind each choice. SsBcmG and wild-type SsBcmE did not crystallize, and the quaternary complex was unattainable with SsBcmC. The perturbed selectivity for SsBcmE T307A (Extended Data Fig. 2) should also be mentioned early. It is not clear how representative its structure would be of its wild-type counterpart. That being said, the ability of this team to overcome challenges in crystallographic characterization is commendable – but the logic needs to be more clearly explained and the disadvantages of using variants needs to be acknowledged.

R: We thank the reviewer for the informative suggestion. We have reorganized the logic of this part according to the comments in the revised manuscript, from line 132 to 185. Given that the revisions are extensive, we will refrain from reiterating them here, but we highlighted the corrected place in the revised manuscript.

(4) On the same note, Supplementary Fig. 6 should include the sequence of SsBcmC, as its tertiary complex structure is solved. In line 118, the words “apo” were used but in a confusing manner; none of the solved structures were free of iron and the co-substrate α KG. For the zoom-in views in Fig. 3, the magenta and blue colors have rendered oxygen and nitrogen atoms hard to see, respectively. It could also be useful to adjust the clipping planes in PyMOL so that the relative depths of residues can be perceived by a reader.

R: We thank the reviewer’s helpful comments. We have added the amino acid sequence of SsBcmC and the Codon-optimized Nucleotide sequence of SobcmC to the Supplementary and added the alignment of SsBcmC to **Supplementary Fig. 6**. We replaced the word “apo” with its corresponding complex name to avoid misunderstanding. **Fig.3** was re-drawn, adding the density map of substrates and adjusting the view of active pockets (this could be found in **Response #Q5**). We also revised the corresponding figure legends.

Revised Supplementary Fig. 6. The sequence alignment of SsBcmE, SoBcmC, SsBcmC and PaBcmG.

(5) In Figure 3, important information is missing that is essential for understanding the key points of the study. Most importantly, an electron density map is shown for the substrate – but the figure caption makes no mention of the type of map or the contour level. Ideally, omit maps should be shown at a minimum of 3.0 sigma. And it would also be optimal show omit maps for the 2OG cosubstrate and any water ligands in the SI. In Fig. 3, I would also recommend showing the metal coordination sphere in the standard orientation with the distal His (higher number) ligand in the lower axial position and the substrate binding site located in the upper axial position. It would also be ideal to show distances between the iron and the nearest carbon atoms in the substrate – inclusion of this information would also help address some of the issues mentioned below.

R: We thank the reviewer for the critical comments. For better understanding and more clearly showing the details of the active site of structures, we re-drew **Fig. 3**, especially the zoom-in part, with the distal His in the lower axial position and the substrate binding site located in the upper axial position. We also added the omit maps of the cofactor α KG and the water coordinated with iron in the **Extended Data Fig. 2**.

Revised Fig. 3. The substrate-bound complex structures reveal detailed active site of Fe(II) and α KG dependent SsBcmE, SoBcmC and PaBcmG. a, b, c, The active pocket of the complex structures of SsBcmE^{T307A}•Fe^{II}• α KG•**1** (green), SoBcmC•Fe^{II}• α KG•**2** (pink) and PaBcmG•Fe^{II}• α KG•**3** (blue), respectively. The above was the overall structure and the below was the detailed interactions of corresponding substrates and residues, shown as sticks. The Fo-Fc (omit map) electron density map (blue mesh) for substrates in the active site is contoured to 3.0 σ . The oxygen atoms were colored in red, nitrogen atoms were colored in blue and the carbon atoms of α KG, **1**, **2**, and **3** were colored in yellow, magenta, cyan, and orange.

Extended Data Fig. 2 The omit map of substrate, α KG, Fe(II) and coordinating water in complex structure of SsBcmE^{T307A}, SoBcmC and PaBcmG. a, b, c, The x-ray structure of complex structures of SsBcmE^{T307A}•Fe^{II}• α KG•**1** (green), SoBcmC•Fe^{II}• α KG•**2** (pink) and PaBcmG•Fe^{II}• α KG•**3** (blue), respectively. The Fo-Fc electron density map (blue mesh) for substrates, α KG, Fe(II) and coordinated water in active site are contoured to 3.0 σ . The oxygen atoms were colored in red, nitrogen atoms were colored in blue and the carbon atoms of α KG, **1**, **2**, and **3** were colored in yellow, magenta, cyan, and orange. The distance of the iron center to the closest carbon atoms or special carbon atom is labeled with black dash line.

(6) In the BcmE T307A crystal structure, C-5a was found to be closest to the iron center among all carbon atoms (line 146, page 9). However, according to Extended Data Fig. 2, BcmE T307A still largely prefers the native selectivity (at C7; product 2) than C-5a (product **1a**), assuming the area ratio under the HPLC traces reflects the molar ratio of molecules. This was attributed to the enlarged pocket that allowed for C4-C5 rotation in the T307A variant (line 148) and could be accounted for with MD simulations. While it is understandable that the dominant conformations do not reflect the actual active conformations (i.e., states that are ready to undergo hydrogen atom transfer), in Extended Data Fig. 4e and f, the active conformations in BcmE T307A seemed more prevalent for C5a than C7, seemingly contradictory to the observed selectivity. Are there any explanations to reconcile these data and computational results? Furthermore, does MD address that why the particular C4-C5 rotamer is preferred in crystallo, bringing C-5a closest to the active site?

R: In BcmE T307A, there were not only C7 hydroxylation products (**2**) but also C5a hydroxylation products (**1a**) (**Extended Data Fig. 3**). DFT calculation showed that the energy barrier of hydrogen transfer of C5a (14.4 kcal/mol) was 1.9 kcal/mol higher than that of C7 (12.5 kcal/mol), indicating that the hydroxylation activity of C5a was lower than that of C7. Therefore, to achieve the hydroxylation of C5a, the proportion of the active conformation of C5a must be significantly higher than that of C7, thus reversing the inherent reactivity. Expectedly, the MD simulation showed that the proportion of the active conformation of C5a was higher, and the crystal structure showed that C5a was found closer to the iron atom, which was consistent with the experimental observation of the hydroxylation of C5a.

(7) Likewise, C-3' was found to be almost equally but slightly closer to the active site than C-2' in the SoBcmC crystal structure (line 165), while C-2' hydroxylation was observed to be the sole outcome (Extended Data Fig. 5). This “contradiction” was again attributed to rotation about the C1'-C2' bond (line 167). If BcmC were to follow the innate reactivity of C-H bonds in the substrate as the authors proposed (Fig. 2b), is substrate positioning still a critical argument (Extended Data Fig. 3b and e) to understand the selectivity of BcmC? If so, how is C-3' accounted for in Extended Data Fig. 3e, and how does MD simulation account for the substrate rotamer observed in the crystal structure? Also, it is desirable to show the activity and selectivity of SsBcmC in Extended Data Fig. 5.

R: We thank the reviewer for the careful and critical comments. In the SoBcmC crystal structure, C-3' was indeed slightly closer to the iron center than C-2', with distances of 5.0 Å and 5.7 Å, respectively. However, this observation should be interpreted in conjunction with the activity data because conformations observed in crystal structures may not accurately represent the active conformations. Notably, we found no hydroxylation products at C-3', a finding that aligns with the absence of an active conformation for C-3' post-MD simulation. We have provided the HPLC result of SsBcmC-catalyzed reaction in the **revised Extended Data Fig. 6** (because of the new **Extended Data Fig. 2.**, all the figure number after it should be plus one). Like SoBcmC,

SsBcmC exhibited high activity (100% yield) and selectivity (100%) for substrate **2**.

Revised Extended Data Fig. 6. HPLC analysis of in vitro reaction of SoBcmC and its mutants.

(8) For PaBcmG, additional minor products (**4a/4b**) were found for wild-type enzyme in Extended Data Fig.6 but not for other variants. Do those minor products exist for SsBcmG, or are they exclusive to PaBcmG? What are the identities of these minor products, as I cannot find any clue throughout the manuscript?

R: Both SsBcmG and PaBcmG could catalyze substrate **3** to produce two byproducts **4a** and **4b**. As described in **question 5** raised by **reviewer #1**, the minor products **4a** and **4b** are two overoxidation diastereomer products that contain an γ,δ -dihydroxy- γ -methyl-proline. In **the revised Extended Data Fig. 7**, we have added the chemical structures of **4a** and **4b** and provided the HPLC result of the SsBcmG-catalyzed reaction.

Revised Extended Data Fig. 7. HPLC analysis of in vitro reaction of PaBcmG and its mutants. The enzymatic reaction (50 μ L) containing 50 mM Tris-HCl buffer (pH 7.5), 0.6 mM **3**, 2 mM α KG, 2 mM L-ascorbic acid, 0.1 mM FeSO₄·7H₂O, and 50 μ g purified enzyme was incubated at 37 $^{\circ}$ C for 1 h. The reactions were quenched by the addition of 100 μ L of pre-cooled methanol and centrifuged at 12,000 rpm for 30 min. The supernatants were analyzed by HPLC. The data show one representative experiment from three independent replicates with similar results. Similar to **4a** and **4b** produced by WT PaBcmG and SsBcmG, the diastereomers **3b** and **3c** are also two overoxidation products. According to our previous research as well as other group studies on BcmG-catalyzed reaction,²⁻⁴ we proposed that PaBcmG-Y288F can catalyze the further oxidation of product **3a** at C3' to produce an aldehyde-containing intermediate, which undergoes addition by amide N2 to form **3b** and **3c**.

References

1. Meng, S.; Han, W.; Zhao, J.; Jian, X.-H.; Pan, H.-X.; Tang, G.-L. A Six-Oxidase Cascade for Tandem C-H Bond Activation Revealed by Reconstitution of

Bicyclomycin Biosynthesis. *Angew. Chem. Int. Ed.* 2018, 57, 719-723.

2. Patteson, J. B.; Cai, W.; Johnson, R. A.; Santa Maria, K. C.; Li, B. Identification of the Biosynthetic Pathway for the Antibiotic Bicyclomycin. *Biochemistry* 2018, 57, 61-65.

3. Witwinowski, J.; Moutiez, M.; Coupet, M.; Correia, I.; Belin, P.; Ruzzini, A.; Saulnier, C.; Caraty, L.; Favry, E.; Seguin, J.; Lautru, S.; Lequin, O.; Gondry, M.; Pernodet, J.-L.; Darbon, E. Study of Bicyclomycin Biosynthesis in *Streptomyces cinnamoneus* by Genetic and Biochemical Approaches. *Sci. Rep.* 2019, 9, 20226.

(9) BcmC selectivity was largely unperturbed by amino-acid replacements around the active site (Extended Data Fig. 5). In contrast, substrate positioning was critical to BcmE and BcmG, as their selectivity was sensitive to replacements of hydrophobic residues (Extended Data Fig. 2) and Y288 (Extended Data Fig. 6), respectively. The authors claimed that these observations can be attributed to three mutually orthogonal strategies separately adopted by the three enzymes (line 323, page 17), but I find that hard to be concluded solely based on the mutagenesis results, especially the sensitivity towards Y308 in BcmE (Extended Data Fig. 2). The authors have also written that “a cumulative set of interactions” (line 318) led to enzymatic selectivity. Is it warranted to only include steric hinderance control as opposed to directing group control for BcmE? On the other hand, I do believe that BcmG invokes directing group control, but it only becomes convincing when the impressive data from non-native substrates are shown (Fig. 5c). I find it speculative (as the authors have also written in line 268) before the strongest evidence is presented, so is it possible to write a section as a last paragraph of the introduction to briefly walk the readers through the logical flow of the manuscript? In addition, in line 290, the neighboring position should be “vicinal” rather than “ortho”, as the functional groups are not part of an aromatic system.

R: We thank the reviewer for the valuable comments. We have added a paragraph at the end of the introduction part to briefly summarize this work as follows. The word

“ortho” was replaced by “vicinal”.

“To reveal the inherent site selectivity of the C(sp^3)–H hydroxylation in the cognate substrates (**1-3**), a truncated catalytic-residue theozyme model was constructed based on the reported spectra data, crystal structure and computational simulations. Our findings revealed that site selectivity for substrates **1** and **3** diverged from that observed in the enzyme catalytic system. We proposed that three α KGDs employ different strategies to control the site selectivity of C–H hydroxylation reactions. Subsequently, we identified the key residues governing selectivity through a combination of crystallographic studies, computational simulations, and biochemical experiments. To validate our hypothesis, we investigated the substrate scope, and the outcomes corroborated our proposed strategies: BcmE, BcmC, and BcmG utilize steric hindrance, inherent reactivity, and a directing group, respectively. Collectively, we elucidated how three α KGDs (BcmE/BcmC/BcmG) employ mutually orthogonal strategies to control the site selectivity of C–H hydroxylation reactions in BCM biosynthesis.”

(10) Finally, the section titles can be more explicit, especially “BcmE and BcmG employ different strategies to control site selectivity compared to BcmC” (line 87). I suggest “BcmE and BcmG defy the innate reactivity preferences of BcmC”, or it becomes too similar to “BcmE, BcmC and BcmG use three different strategies to control the site selectivity of C-H oxidation” (line 189).

R: We took the reviewer’s suggestion to change the section title to “BcmE and BcmG defy the innate reactivity preferences of BcmC”.

Minor comments:

(1) Page 4, line 60: “ α KGDs amino acid hydroxylases”. Since α KGDs are already

defined as “non-heme FeII/ α -ketoglutarate-dependent dioxygenase” (page 3, line 54), consider “ α KG-dependent amino acid hydroxylases”.

R: We have corrected it to “ α KG-dependent amino acid hydroxylases”.

(2) Page 4, line 65: “homologous enzymes that catalyzed C-H hydroxylation”. Should be “catalyze”.

R: We have corrected “catalyzed” to “catalyze”.

(3) Fig. 1b1, b2, b3 figure titles: “homolog enzyme”. Should be “homologous enzymes”. Similarly for the figure captions.

R: We have changed all “homolog enzyme” to “homologous enzymes” in the figure titles of Fig.1b1-3 and figure captions.

(4) Page 5, line 78: “In our previously biosynthetic studies”. Should be “previous”.

R: The word “previously” has been corrected as “previous”.

(5) Page 5, starting line 90: The incorporation of a truncated theozyme is an interesting way to quantify the intrinsic reactivity, but it took me a while to realize its importance. Can the readers be more explicit on: i. theozyme only considers primary coordination sphere such that the substrate could possibly be accessed from all orientations by the iron center, ii. inherent site selectivity is based on bond dissociation energy (cite Supplementary Fig. 1).

R: We thank the reviewer for the valuable advice. To reveal the inherent site selectivity of the C(sp^3)-H hydroxylation in the cognate substrates (**1-3**), a truncated catalytic-residue theozyme model was constructed based on the reported spectra data, crystal structure and computational simulations (*Nat. Commun.* **9**, 1168 (2018); *J. Am. Chem. Soc.* **140**, 7116-7126 (2018); *Phys. Chem. Chem. Phys.* **19**, 20188-20197 (2017); *Nat. Catal.* **6**, 637-648 (2023)). In detail, this model contained a cognate substrate, Fe^{IV}-oxo species, two methylimidazoles for two histidine residues, two acetate anions in aspartic

acid and succinate, as well as one water. The theozyme model only considered the primary coordination sphere such that the substrate could be accessed from all orientations by the iron center, to quantitatively evaluate the inherent reactivity of the C(sp^3)-H bond by using the hydrogen transfer energy barrier of the substrate.

(6) Fig. 2a: How is this ferryl orientation (i.e., trans to this particular histidine) chosen?

What is the evidence of a water ligand?

R: Previous spectra data, crystal structure and computational simulations on the related α KGD indicated that a succinate-coordinated Fe^{IV}-oxo complex is the reactive species (*Nat. Commun.* **9**, 1168 (2018); *J. Am. Chem. Soc.* **140**, 7116-7126 (2018); *Phys. Chem. Chem. Phys.* **19**, 20188-20197 (2017); *Nat. Catal.* **6**, 637-648 (2023)). Based on these studies, a truncated catalytic-residue theozyme model containing a cognate substrate, Fe^{IV}-oxo species, two methylimidazoles for two histidine residues, two acetate anions in aspartic acid and succinate, as well as one water, was constructed.

(7) Fig. 2b title: “different types of hydroxylation (kcal/mol)”. Should be “hydrogen abstraction”, unless the subsequent oxygen rebound step is considered.

R: The “hydroxylation” has been corrected to “hydrogen abstraction” in the revised manuscript.

(8) Fig. 2b caption title: “Gibbs free energies ... for the hydroxylation reactions”. Consider “Gibbs free energy barriers ... for the hydrogen abstraction reactions”. Also, there needs to be an ending parenthesis after “iron: cyan”. “The energy barriers of every C-H bond were” mixes plural and singular forms.

R: The “hydroxylation” has been corrected to “hydrogen abstraction”. The ending parenthesis after “iron: cyan” has been completed. “The energy barriers of every C-H bond were” has been corrected to “The energy barrier of every C-H bond was” in the revised manuscript.

(9) Page 7, line 119: what is the purpose of italicizing “X” in “HXD-Xn-H”?

R: We are sorry that the italicizing “X” in “HXD-Xn-H” was a clerical error without any meaning. We have corrected it in the revised manuscript.

(10) Page 7, line 122: “r.m.s.d” should be “r.m.s.d.”, and there are too many significant digits as the resolution should not allow for accuracy to the third decimal place. Consider two decimal places throughout the manuscript.

R: We have corrected it to “r.m.s.d.”, and all digits of resolution changed to two decimal throughout the manuscript.

(11) Fig.3 caption title: “FeII and α KG dependent”. Consider “Fe(II)- and α KG-dependent”. In addition, “megentas”, “cyans”, and “oranges” should be singular and please correct the spelling of magenta.

R: The cation title of **Fig. 3** was replaced by “Fe(II)- and α KG-dependent”, and all spelling mistakes were corrected.

(12) Page 8, line 140: it seems odd to characterize the interaction between carbonyl and F273 as π - π stacking.

R: We have corrected the sentence to “The carbonyl group of the amide part of the DKP ring is stabilized by a π - π stacking interactions with the aromatic ring F273.” in the revised manuscript, and this referred from our previous work (*Nature Catalysis*, 6, 2023, 637-648.).

(13) Page 8, line 142: no need to write “respectively” in this sentence.

R: We have deleted the word “respectively”.

(14) Page 9, line 150: “wildtype” should be “wild-type”.

R: We have replaced all “wildtype” with “wild-type”.

(15) Page 9, line 151: “T307 could pull C7 directing to the iron center”. Consider “could bring C7 closer to the iron center”.

R: We have replaced “T307 could pull C7 directing to the iron center” with “T307 could bring C7 closer to the iron center”.

(16) Page 9, lines 152-153: “Residues M177, portion of 1.” Could cite Fig. 3a here.

R: We cited **Fig. 3a** after “Residues M177, ... portion of 1.”

(17) Page 9, line 157: “we infer ...”. This language seems a bit too strong without the upcoming mutagenesis studies. Maybe “hypothesize” is a better word.

R: The word “infer” was replaced by “hypothesize”.

(18) Page 10, line 178: is S175 not shown in Fig. 3c?

R: S175 was shown and labeled in the revised **Fig. 3c**.

(19) Page 10, line 179: no need to write “respectively” in this case.

R: We have deleted the word “respectively”.

(20) Page 11, line 197: “three enzymes”. Consider “these enzymes” or “these three enzymes”.

R: We have added the word “these” before “three enzymes”.

(21) Page 14, line 244: “easily reactive”. Consider “inherently more reactive”.

R: We changed “easily reactive” to “inherently more reactive”.

(22) Lines 470 and 484, page 23: “was/were stored in our laboratory”. Consider “was/were previously reported by our laboratory”.

R: We have followed your suggestion and modified it to “Compounds **1–4**, **1b**, **1d**, **4a**, and **4b** were previously reported by our laboratory”.

(23) Extended Data Figs. 2, 5, and 6: it might be useful to number the modified carbons

for the molecular structures.

R: We numbered the modified carbon for the molecular structures, please see the revised **Extended Data Fig. 3, 6, and 7** (for adding a new **Extended Data Fig.2**, the later figure number plus one).

(24) Extended Data Table 1: “C121” and “P121” should be “C21” and “P21”, respectively.

R: We have taken the suggestion and changed the writing forms of the space group in the revised **Extended Data Table 1**.

(25) Supplementary Fig. 1: “viarous” should be “various”.

R: We have corrected the misspelling word as “various”.

(26) Supplementary Fig. 5d: “C-termonal” should be “C-termini”.

R: We have corrected the typing errors, and the revised **Supplementary Fig. 5d** as follows.

Revised Supplementary Fig. 5. The structure of BcmE, BcmC and BcmG without substrates.

(27) Supplementary Fig. 5b figure caption: the use of “2OG” is not consistent with the text (“ α KG”).

R: We replaced the “2OG” to “ α KG” in **Supplementary Fig. 5b** figure caption, as follows “Cosubstrate α KG was shown as yellow sticks”.

(28) Supplementary Fig. 5c figure caption: “palecyan” should be “pale cyan”.

R: We have corrected this word as “colored in pale cyan” in the revised caption of **Supplementary Fig. 5c**.

(29) Supplementary Fig. 5d figure caption: “the coarser the higher”. Consider “the thicker the higher”.

R: We have changed the word “coarser” to “thicker”, as follows “The thickness of putty is directly related to the b-factor, the thicker the higher”.

(30) Supplementary Fig. 7 figure caption: “magentas” should be “magenta” and “palecyan” should be “pale cyan”.

R: We have changed the errors in the revised caption of **Supplementary Fig. 7** as “**b, d, f** were the alignment of apo BcmE (AF2 model, grey)/BcmC (magenta)/BcmG (pale cyan) and their corresponding complex structures”

Reviewer #3 (Remarks to the Author):

In this work a crystallography and computational study is reported on understanding of three nonheme iron dioxygenases that react with the same substrate but give different product distributions. They trap three crystals of structures of the three isozymes and do MD simulations to understand the reactivity differences. Also substrate analogues are tested that give further insight into the C-H activation pathways. Overall it is

interesting work that fits the remit of Nature Communications.

1. It is generally accepted that the second-coordination sphere in nonheme iron dioxygenases determines substrate binding and selectivity. This will be useful to mention here and give appropriate citations. I find it difficult to see from Figure 3 what the difference in second coordination sphere is between the various structures. What determines the substrate binding and positioning?

R: Thanks a lot for your professional and helpful comment. We added a sentence to the revised manuscript as follows, “**We specially investigated the residues in the second coordinate sphere of these three enzymes and found out it was consistent with previous research, that the second coordinate sphere determining the substrate recognition and selectivity.**” and referred the corresponding literatures (*Acc. Chem. Res.* **55**, 2480-2490 (2022); *Bulletin of the Chemical Society of Japan* **97**, doi:10.1093/bulcsj/bcsj.20230224 (2023); *Metallomics* **5**, 287-301 (2013); *Biochemistry* **42**, 7294-7302 (2003)). We also re-drew the **Fig. 3**, it could be found as follow (**Revised Fig. 3**). Ignoring the results of the activity experiments, the active sites of the three enzymes appeared to have some similar steric hindrance and hydrogen bonding interactions. However, when combined with activity assays, it became clear which interactions were decisive for selectivity and which were not such significant. We discussed these key residues in the manuscript. For BcmE, a π - π stacking interactions between DKP ring of **1** with the aromatic ring F273 and steric hindrance of T307 might play a greater role in substrate binding and position (**Fig.3a, Extended Data Fig. 2a and Extended Data Fig. 4a**). For BcmC, T170, N277 and W281 may the key residues impacting the substrate binding and positioning. For BcmG, we found Y288 might be the key residues determining the substrate binding and positioning.

Revised Fig. 3. The substrate-bound complex structures reveal detailed active site of Fe(II) and α KG dependent SsBcmE, SoBcmC and PaBcmG.

2. The authors start with a set of minimal cluster DFT studies on the first coordination sphere only. It has been shown by several groups that these small cluster models often give the wrong energetics and selectivity (the authors may want to mention this), and indeed the energies in Figure 2 show that the DFT does not predict the selectivity correctly. It would have been better if large DFT clusters had been used based on the new crystal structures. Do these larger structures give insight into the selectivity of the enzymes?

R: QM/MM calculations or DFT calculations including the active site residues are primarily intended to explain the underlying principles and controlling factors of the protein scaffold for reactivity and selectivity, but are different from our aim of resolving the intrinsic site selectivity of the substrate.

To reveal the inherent site selectivity of the $C(sp^3)$ -H hydroxylation in the cognate substrates (**1-3**), a truncated catalytic-residue theozyme model was constructed based on the reported spectra data, crystal structure and computational simulations (*Nat. Commun.* **9**, 1168 (2018); *J. Am. Chem. Soc.* **140**, 7116-7126 (2018); *Phys. Chem. Chem. Phys.* **19**, 20188-20197 (2017); *Nat. Catal.* **6**, 637-648 (2023)). In detail, this model contained a cognate substrate, Fe^{IV} -oxo species, two methylimidazoles for two histidine residues, two acetate anions in aspartic acid and succinate, as well as one water. The theozyme model only considered primary coordination sphere such that the substrate could possibly be accessed from all orientations by the iron center, so as to quantitatively evaluate the inherent reactivity of the $C(sp^3)$ -H bond by using the hydrogen transfer energy barrier of the substrate. As pointed out by **Reviewer 2#**, "the incorporation of a truncated theozyme is an interesting way to quantify the intrinsic reactivity. This theozyme model only considers primary coordination sphere such that the substrate could possibly be accessed from all orientations by the iron center. And the inherent site selectivity is based on bond dissociation energy (**Supplementary Fig. 1**)."

Similarly, DFT calculations based on theozyme models have been used to reveal the inherent size selectivity in ring-closure reactions as well as regioselectivity and periselectivity among pericyclic reactions, etc., as detailed in our previous work and in K.N. Houk's articles (*Nat. Catal.* **6**, 637-648 (2023); *Nature* **586**, 64-69 (2020); *Nat. Catal.* **4**, 1059-1069 (2021); *Nat. Catal.* **4**, 223-232 (2021)).

Supplementary Fig. 1. The bond dissociation energy of different type of C-H.

3. Is the polarity different in the active site pockets of the three enzymes?

R: The polarity of the three enzymes is different in the active site pocket. BcmE and BcmG are more polar than BcmC. We could find out this difference from the comparison of surface charge of three different proteins.

Fig. R3 The charge surface of the active pocket of complex structures of **SsBcmE^{T307A}** (a), **SoBcmC** (b) and **PaBcmG** (c). Red and blue color represent negative and positive charge, respectively. The more polar the color is more dark (red or blue). Grey or white presents close to hydrophobic.

4. How do the structure of the pdb's compare with other pdb's in the protein databank?

R: We conducted a structural comparison using the Dali server ([Dali server \(helsinki.fi\)](http://dali.helsinki.fi)), which revealed that the closest match was the structure 7V3N (protein

BcmB, part of the bicyclomycin biosynthesis pathway), with a root-mean-square deviation (r.m.s.d.) value ranging from 0.857 to 1.047 Å (the details was shown in **Fig. R4**). BcmB is a multifunctional α KGD that uses BcmG-catalyzed product **3** as the substrate to selectively construct the oxa-bridged eight-membered ring in bicyclomycin biosynthesis through sequential dehydrogenation, epoxidation and ring-closure reactions. (*Angew. Chem. Int. Ed.* **57**, 719 –723 (2018).; *Nat. Catal.* **6**, 637–648 (2023).). Among the top-ranked structures in PDB25 or PDB50, there are Fe^{II}/ α KGD dependent hydroxylation dioxygenases such as EFes and H6H. Most of the corresponding r.m.s.d. values are high, which indicating significant structural variation. However, exceptions exist, with BcmC and 5O7Y (an EFE) showing notably higher similarity to each other (**Fig. R4**).

Fig. R4 The structural comparison of three complex with 5V2Z (dark grey, H6H) and 5O7Y (light grey, EFE). Three complex structure were SsBcmE^{T307A}.

Fe^{II}•αKG•**1** (green, 8HXY), SoBcmC•Fe^{II}•αKG•**2** (pink, 8XHQ) and PaBcmG•Fe^{II}•αKG•**3** (blue, 8XHX).

5. How do the DFT calculations compare with DFT and QM/MM studies from the literature? There is a lot of work in the literature by Christov, de Visser, Borowski, Gauld and others.

R: QM/MM calculations or DFT calculations including the active site residues are more likely to be used to analyze the underlying principles and controlling factors of enzyme microenvironment on the activity and selectivity (*ACS Catal.* **10**, 1195-1209 (2020)).

The truncated catalytic-residue theozyme model only considered primary coordination sphere such that the substrate could possibly be accessed from all orientations by the iron center, and the energy barrier obtained by DFT calculation can quantitatively evaluate the inherent reactivity of the C(*sp*³)-H bond the substrate. As pointed out by **reviewer 2**, “the incorporation of a truncated theozyme is an interesting way to quantify the intrinsic reactivity. This theozyme model only considers primary coordination sphere such that the substrate could possibly be accessed from all orientations by the iron center. And the inherent site selectivity is based on bond dissociation energy (Supplementary Fig. 1).” Further, DFT calculation based on the theozyme model and MD simulations can be used to predict the selectivity of enzymes (*Chem. Catalysis* **2**, 2658-2674 (2022)).

And we added some references of Christov, de Visser, Borowski, Gauld in Methods part of our revised manuscript as follow.

60. Rifayee, S. et al. Catalysis by KDM6 Histone Demethylases - A Synergy between the Non-Heme Iron(II) Center, Second Coordination Sphere, and Long-Range Interactions. *Chemistry* **29**, e202301305 (2023).

61. Chaturvedi, S. S. et al. Catalysis by the Non-Heme Iron(II) Histone Demethylase PHF8 Involves Iron Center Rearrangement and Conformational Modulation of Substrate Orientation. *ACS Catal.* **10**, 1195-1209 (2020).

62. Yeh, C. C. G. et al. Cluster Model Study into the Catalytic Mechanism of α -Ketoglutarate Biodegradation by the Ethylene-Forming Enzyme Reveals Structural Differences with Nonheme Iron Hydroxylases. *ACS Catal.* **12**, 3923-3937 (2022).
63. Chaturvedi, S. S. et al. Dioxygen Binding Is Controlled by the Protein Environment in Non-heme Fe(II) and 2-Oxoglutarate Oxygenases: A Study on Histone Demethylase PHF8 and an Ethylene-Forming Enzyme. *Chemistry* **29**, e202300138 (2023).
64. Aboelnga, M. M. & Gauld, J. W. Establishing a substrate-assisted mechanism for the pre-transfer editing in SerRS and IleRS: a QM/QM investigation. *Structural Chemistry* **35**, 519-530 (2024).
65. Ali, H. S., Warwicker, J. & de Visser, S. P. How Does the Nonheme Iron Enzyme NapI React through l-Arginine Desaturation Rather Than Hydroxylation? A Quantum Mechanics/Molecular Mechanics Study. *ACS Catal.* **13**, 10705-10721 (2023).
66. de Visser, S. P., Mukherjee, G., Ali, H. S. & Sastri, C. V. Local Charge Distributions, Electric Dipole Moments, and Local Electric Fields Influence Reactivity Patterns and Guide Regioselectivities in α -Ketoglutarate-Dependent Non-heme Iron Dioxygenases. *Accounts of Chemical Research* **55**, 65-74 (2022).
67. Hardy, F. J. et al. Probing Ferryl Reactivity in a Nonheme Iron Oxygenase Using an Expanded Genetic Code. *ACS Catal.* **14**, 11584-11590 (2024).
68. Waheed, S. O. et al. Role of Structural Dynamics in Selectivity and Mechanism of Non-heme Fe(II) and 2-Oxoglutarate-Dependent Oxygenases Involved in DNA Repair. *ACS Cent Sci* **6**, 795-814 (2020).
69. Thomas, M. G. et al. The Unique Role of the Second Coordination Sphere to Unlock and Control Catalysis in Nonheme Fe(II)/2-Oxoglutarate Histone Demethylase KDM2A. *Inorg Chem* **63**, 10737-10755 (2024).
101. Álvarez-Barcia, S. & Kästner, J. Atom Tunneling in the Hydroxylation Process of Taurine/ α -Ketoglutarate Dioxygenase Identified by Quantum Mechanics/Molecular Mechanics Simulations. *J. Phys. Chem. B* **121**, 5347-5354 (2017).

6. The MD simulations seem to imply that substrate is at quite a large distance from the

oxidant, which may be a problem for catalysis. This may require some discussion.

R: MD simulation analyzed the active conformation ratio of different C-H bond hydroxylation of the substrate to predict the site selectivity of the enzyme.

As shown in **Extended Data Fig. 4-5**: (i) For the hydroxylation of substrate **1** catalyzed by SsBcmE, the hydrogen atoms on other carbons were far away from the oxygen atom in the Fe^{IV}-oxo species, and only the hydrogen atoms on C7 were closest to the Fe^{IV}-oxo species. The active conformation existed and the ratio was 10/5000 (the conformation meets the distance between the O-H to be formed $\leq 2.8 \text{ \AA}$ and the angle among the O-H-C is $170 \pm 15^\circ$), suggesting that the product may be mainly hydroxylated by C7.

(ii) For the hydroxylation of substrate **2** catalyzed by SoBcmC, the hydrogen atoms on other carbons were far away from the oxygen atoms in the Fe^{IV}-oxo species, only the hydrogen atom on C2' was closest to the Fe^{IV}-oxo species, and the active conformation ratio was 122/5000, suggesting that the product may be mainly hydroxylated by C2'.

(iii) For the hydroxylation of PaBcmG catalyzed substrate **3**, the hydrogen atoms on other carbons were far away from the oxygen atoms in the Fe^{IV}-oxo species, only the hydrogen atoms on C3' were closest to the Fe^{IV}-oxo species, and the active conformation ratio was 327/5000, suggesting that the product may be mainly hydroxylated by C3'.

(iv) For the hydroxylation of substrate **1** catalyzed by SsBcmE^{F273A}, the hydrogen atoms on other carbons were far away from the oxygen atoms in the Fe^{IV}-oxo species, and only the hydrogen atoms on C6 were closest to the Fe^{IV}-oxo species, and the active conformation ratio was 49/5000, suggesting that the product may be mainly hydroxylated by C6. It is worth mentioning that the site selectivity of enzyme predicted by calculation was consistent with that measured by experiment.

Reviewer #4 (Remarks to the Author):

References

1. Dunham, N. P.; Chang, W. C.; Mitchell, A. J.; Martinie, R. J.; Zhang, B.; Bergman, J. A.; Rajakovich, L. J.; Wang, B.; Silakov, A.; Krebs, C.; Boal, A. K.; Bollinger, J. M., Jr., Two Distinct Mechanisms for C-C Desaturation by Iron(II)- and 2-(Oxo)glutarate-Dependent Oxygenases: Importance of alpha-Heteroatom Assistance. *J. Am. Chem. Soc.* **140** (23), 7116-7126 (2018).
2. Meng, S.; Han, W.; Zhao, J.; Jian, X. H.; Pan, H. X.; Tang, G. L., A Six-Oxidase Cascade for Tandem C-H Bond Activation Revealed by Reconstitution of Bicyclomycin Biosynthesis. *Angew Chem. Int. Ed.* **57** (3), 719-723 (2018).
3. Patteson, J. B.; Cai, W.; Johnson, R. A.; Maria, K. C. S.; Li, B., Identification of the Biosynthetic Pathway for the Antibiotic Bicyclomycin. *Biochemistry*, **57** (1), 61-65 (2018).
4. Witwinowski, J.; Moutiez, M.; Coupet, M.; Correia, I.; Belin, P.; Ruzzini, A.; Saulnier, C.; Caraty, L.; Favry, E.; Seguin, J.; Lautru, S.; Lequin, O.; Gondry, M.; Pernodet, J.-L.; Darbon, E., Study of bicyclomycin biosynthesis in *Streptomyces cinnamoneus* by genetic and biochemical approaches. *Sci. Rep.* **9**, 20226 (2019).
5. Chakrabarty, S.; Wang, Y.; Perkins, J. C.; Narayan, A. R. H., Scalable biocatalytic C-H oxyfunctionalization reactions. *Chem. Soc. Rev.* **49** (22), 8137-8155 (2020).
6. Münch, J.; Püllmann, P.; Zhang, W.; Weissenborn, M. J., Enzymatic Hydroxylations of sp³-Carbons. *ACS Catal.* **11** (15), 9168-9203 (2021).
7. Ushimaru, R.; Ruszczycky, M. W.; Chang, W.-c.; Yan, F.; Liu, Y.-n.; Liu, H.-W., Substrate Conformation Correlates with the Outcome of Hyoscyamine 6 beta-Hydroxylase Catalyzed Oxidation Reactions. *J. Am. Chem. Soc.* **140** (24), 7433-7436 (2018).
8. Cao, Y.; Hay, S.; de Visser, S. P., An Active Site Tyr Residue Guides the

Regioselectivity of Lysine Hydroxylation by Nonheme Iron Lysine-4-hydroxylase Enzymes through Proton-Coupled Electron Transfer. *J. Am. Chem. Soc.* **146** (17), 11726-11739 (2024).

9. Bollinger, J. M., Jr.; Krebs, C.; Boal, A. K.; Silakov, A.; Price, J. C.; Matthews, M. L.; Chang, W.-C.; Martinie, R. J.; Pan, J.; Dunham, N. P.; Wenger, E.; Copeland, R. A.; Lin, C.-Y., Progress Toward Understanding Protein Control of Reaction Outcome in the Diverse Reactivity of Iron(II)- and 2-Oxoglutarate-dependent Oxygenases. *Faseb J.* **36**. <https://doi.org/10.1096/fasebj.2022.36.S1.0I227> (2022).

10. Karamzadeh, B.; Kumar, D.; Sastry, G. N.; de Visser, S. P., Steric Factors Override Thermodynamic Driving Force in Regioselectivity of Proline Hydroxylation by Prolyl-4-hydroxylase Enzymes. *J. Phys. Chem. A* **114** (50), 13234-13243 (2010).

11. Helmetag, V.; Samel, S. A.; Thomas, M. G.; Marahiel, M. A.; Essen, L.-O., Structural basis for the erythro-stereospecificity of the L-arginine oxygenase VioC in viomycin biosynthesis. *Febs. J.* **276** (13) 3669-3682 (2009).

12. Matthews, M. L.; Neumann, C. S.; Miles, L. A.; Grove, T. L.; Booker, S. J.; Krebs, C.; Walsh, C. T.; Bollinger, J. M., Jr., Substrate positioning controls the partition between halogenation and hydroxylation in the aliphatic halogenase, SyrB2. *Proc. Natl. Acad. Sci. U.S.A.*, **106** (42), 17723-17728 (2009).

13. Müller, I.; Stückl, C.; Wakely, J.; Kertesz, M.; Usón, I., Succinate complex crystal structures of the α -ketoglutarate-dependent dioxygenase AtsK -: Steric aspects of enzyme self-hydroxylation. *J. Biol. Chem.* **280** (7), 5716-5723 (2005).

14. Ali, H. S.; Henchman, R. H.; de Visser, S. P., What Determines the Selectivity of Arginine Dihydroxylation by the Nonheme Iron Enzyme OrfP? *Chem. Eur. J.* **27** (5), 1795-1809 (2021).

15. Di Giuro, C. M. L.; Konstantinovic, C.; Rinner, U.; Nowikow, C.; Leitner, E.; Straganz, G. D., Chiral Hydroxylation at the Mononuclear Nonheme Fe(II) Center of 4-Hydroxymandelate Synthase - A Structure-Activity Relationship Analysis. *Plos One* **8** (7) (2013).

16. Hangasky, J. A.; Taabazuing, C. Y.; Martin, C. B.; Eron, S. J.; Knapp, M. J., The facial triad in the α -ketoglutarate dependent oxygenase FIH: A role for sterics in

- linking substrate binding to O₂ activation. *J. Inorg. Biochem.* **166**, 26-33 (2017).
17. Wojdyla, Z.; Borowski, T., Properties of the Reactants and Their Interactions within and with the Enzyme Binding Cavity Determine Reaction Selectivities. The Case of Fe(II)/2-Oxoglutarate Dependent Enzymes. *Chem. Eur. J.* **28**, e202104106 (2022).
 18. de Visser, S. P., Second-Coordination Sphere Effects on Selectivity and Specificity of Heme and Nonheme Iron Enzymes. *Chem. Eur. J.* **26** (24), 5308-5327 (2020).
 19. de Visser, S. P., What factors influence the ratio of C-H hydroxylation versus C=C epoxidation by a nonheme cytochrome P450 biomimetic? *J. Am. Chem. Soc.* **128** (49), 15809-15818 (2006).
 20. Meng, G.; Lam, N. Y. S.; Lucas, E. L.; Saint-Denis, T. G.; Verma, P.; Chekshin, N.; Yu, J.-Q., Achieving Site-Selectivity for C-H Activation Processes Based on Distance and Geometry: A Carpenter's Approach. *J. Am. Chem. Soc.* **142** (24), 10571-10591 (2020).
 21. Cochrane, R. V. K.; Vederas, J. C., Highly Selective but Multifunctional Oxygenases in Secondary Metabolism. *Acc. Chem. Res.* **47** (10), 3148-3161 (2014).
 22. Mukherjee, G.; Satpathy, J. K.; Bagha, U. K.; Mubarak, M. Q. E.; Sastri, C. V.; de Visser, S. P., Inspiration from Nature: Influence of Engineered Ligand Scaffolds and Auxiliary Factors on the Reactivity of Biomimetic Oxidants. *Acs Catal.* **11** (15), 9761-9797 (2021).
 23. White, P. W., Mechanistic studies and selective catalysis with cytochrome P-450 model systems. *Bioorg. Chem.* **18** (4), 440-456 (1990).
 24. Lundberg, M.; Borowski, T., Oxoferryl species in mononuclear non-heme iron enzymes: Biosynthesis, properties and reactivity from a theoretical perspective. *Coord. Chem. Rev.* **257** (1), 277-289 (2013).
 25. de Beer, S. B. A.; van Bergen, L. A. H.; Keijzer, K.; Rea, V.; Venkataraman, H.; Guerra, C. F.; Bickelhaupt, F. M.; Vermeulen, N. P. E.; Commandeur, J. N. M.; Geerke, D. P., The Role of Protein Plasticity in Computational Rationalization Studies on Regioselectivity in Testosterone Hydroxylation by Cytochrome P450 BM3 Mutants. *Curr. Drug Metab.* **13** (2), 155-166 (2012).
 26. Johnston, J. B.; Ouellet, H.; Podust, L. M.; de Montellano, P. R. O., Structural

control of cytochrome P450-catalyzed ω -hydroxylation. *Arch. Biochem. Biophys.* **507**
(1), 86-94 (2011).

Responses to Reviewers' comments

We are submitting our revised manuscript (NCOMMS-24-45512-A) entitled “**Three distinct strategies lead to programmable aliphatic C–H oxidation in bicyclomycin biosynthesis**” back to *Nature communication*. We sincerely appreciate all the comments from the reviewers, which have been invaluable in enhancing the quality of our manuscript.

We have revised our manuscript carefully and addressed all the comments and concerns point-to-point. In summary, we have made the following changes: 1) added **Fig. R2** to the **Extended Data Fig. 7**; 2) included a summary statement as suggested by the reviewer; 3) removed a sentence that deemed unworthy of comment; 4) provided the relevant product yields; 5) discussed the results of the BcmE mutation activity assay in conjunction with the corresponding structural and MD simulation results.

For the reviewer and editor's convenience, we have uploaded both the clean copy and the track version highlighting changes in **green background** of main text and SI.

Our point-by-point responses to the reviewers is included below in this file.

Remarks from the reviewers are shown in black. Our responses are shown in **blue**.

Reviewer #1 (Remarks to the Author):

The authors have satisfactorily addressed this reviewer's comments, and the manuscript is recommended for acceptance with one suggestion: Figure R2 should be included in the manuscript as a supplementary figure.

R: We thank the Reviewer and take the suggestion to add **Figure R2** into the revised manuscript as **Extended Data Fig. 7**.

Revised Extended Data Fig. 7 HPLC analysis of in vitro reaction of PaBcmG and its mutants.

Reviewer #2 (Remarks to the Author):

The authors have submitted a revision that addresses prior critiques. The authors have devoted an introductory paragraph for an overview of the logical framework in this manuscript – although this paragraph could include more details to help the reader understand exactly which experiments or results were key in defining the proposed reaction mechanisms.

A few remaining questions include the major and minor issues outlined below.

In the description of the theozyme (line 109), why is a water ligand included in the model of the reactive intermediate? The last sentence (lines 115-118) in this section is not comprehensible. I think

a summary statement is needed to sum up the findings from the computational assessment. It seems that for all enzymes evaluated, the secondary carbon targets yield lower energy barriers – and this is sensible because target of this position by HAT would result in a more stable radical. Then state that BcmC targets the most reactive position but BcmE and BcmG react with sites that are intrinsically less reactive. But I think it should be mentioned here that the comparisons made (C2' versus C7 and C5 versus C3') are on completely different sides of the molecule. And it is common knowledge that Fe/2OG, P450, and many other enzymes that employ reactive iron intermediates use substrate positioning to target specific sites that may not be the most reactive on a given substrate. I think the relevant literature precedent should be cited here because this is not a new idea.

R: Thanks! Based on these reported spectroscopic experiments and computational results (*Nat. Commun.* 2018, 9, 1168; *Phys. Chem. Chem. Phys.* 2017, 19, 20188–20197; *J. Am. Chem. Soc.* 2018, 140, 7116–7126), we constructed the theozyme model with a water as a ligand. The binding of water molecules leads to the formation of a reactive Fe^{IV}-oxo species, which then have its oxo group attack the substrate's C-H bond. This is followed by an OH rebound to produce a hydroxylated product.

To better understand the last sentence (previous lines 115-118), we took the reviewer's suggestion to add a summary statement for revision: "As expected, substrate 1-3, catalyzed by a theozyme without the protein scaffold, has the highest activity of tertiary carbon, C2', C2', and C5, respectively. In fact, BcmC targets the most reactive position (C2') but BcmE and BcmG react with sites that are intrinsically less reactive (C7 not C2'; C3' not C5), indicating that the specific enzyme scaffold is another essential factor influencing the regioselectivity of the substrate^{39,45,47,65,77-79}."

And we also added some relative references as follow:

39. Elijah N. Kissmana et al. Biocatalytic control of site-selectivity and chain lengthselectivity in radical amino acid halogenases. *Proc. Natl. Acad. Sci. U.S.A.* **120**, e2214512120 (2023).

45. Ali, H. S. & de Visser, S. P. Catalytic divergencies in the mechanism of L-arginine hydroxylating nonheme iron enzymes. *Front. Chem.* **12**, 1365494 (2024).

47. Ali, H. S., Henchman, R. H., Warwicker, J. & de Visser, S. P. How Do Electrostatic Perturbations of the Protein Affect the Bifurcation Pathways of Substrate Hydroxylation versus Desaturation in the Nonheme Iron-Dependent Viomycin Biosynthesis Enzyme? *J. Phys. Chem. A* **125**, 1720-1737 (2021).

65. Ali, H. S., Warwicker, J. & de Visser, S. P. How Does the Nonheme Iron Enzyme NapI React through l-Arginine Desaturation Rather Than Hydroxylation? A Quantum Mechanics/Molecular Mechanics Study. *ACS Catal.* **13**, 10705-10721 (2023).

77. Wenger, E. S. et al. Optimized Substrate Positioning Enables Switches in the C-H Cleavage Site and Reaction Outcome in the Hydroxylation-Epoxidation Sequence Catalyzed by Hyoscyamine 6 β -Hydroxylase. *J. Am. Chem. Soc.* **146**, 24271-24287 (2024).

78. Mitchell, A. J. Structural Basis for Alternative Reaction Outcome in Iron and 2-Oxoglutarate-Dependent Oxygenases. (2017).

79. Karamzadeh, B., Kumar, D., Sastry, G. N. & de Visser, S. P. Steric Factors Override Thermodynamic Driving Force in Regioselectivity of Proline Hydroxylation by Prolyl-4-hydroxylase Enzymes. *J. Phys. Chem. A* **114**, 13234-13243 (2010).

The sentences in lines 195-199 do not make sense. The first sentence states that binding substrate

yields “a more compact active site” but the overlay of “apo” and substrate bound structures in figure S7d does not show any significant conformational changes in the overall fold. It does appear that the C-terminus of the protein might order to cover up the active site- but I would not call that a compaction of the active site – it is simply covering it up or shielding from solvent. Additionally, the authors speculate that this conformational change that occurs on binding of substrate somehow promotes dissociation of a water ligand and binding of O₂ to the iron cofactor. However, the complex with substrate still shows a water ligand bound. I don’t think this is worthy of comment. Additionally, the appropriate references for this mechanism of substrate triggered O₂ addition are not included. Some review articles are cited but the original research articles that give rise to this mechanism previously proposed by others are not referenced.

R: We thank the reviewer for the critical comments and decide to remove this part (previous line 195-199) from the original manuscript.

In the section starting on page 12, it would be helpful to include information about yield when relevant. For example, the reactions of BcmE with variants that dramatically alter regioselectivity (giving 1b/c products) also show lesser amounts of product accumulation. It would also be useful to include more structural context for these results. Why do the authors think that the other side of the molecule can be targeted when certain mutations are made to open up the active site? In general this section would benefit from adding more context about how the mutagenesis and MD results fit in with the previously discussed structural work.

R: We thank the reviewer very much for the thoughtful and valuable comments. We have added the relevant information about yield in the revised manuscript (from line 244 to line 309). To analyze the factors that contribution to the generation of products **1a/b/c**, we performed MD simulations with mutants T307A, T307L and F273A, respectively. As expected, the dominant active conformations for hydroxylation in the F273A, T307L and T307A systems are C6, C2' and C5a (shown in the dotted rectangle: 49/5000, 90/5000 and 14/5000, Extended Data Fig. 5b, d and f), respectively, which was consistent with our mutagenesis results. In detail, substrate **1** binds to the active sites of these three mutant enzymes in different binding patterns, with each pre-reaction site oriented toward the active Fe^{IV}-oxo species for subsequent hydrogen atom abstraction, which was consistent with our structure, particularly the crystal structure of SsBcmE^{T307A}•Fe^{II}•αKG•**1**.

To better understand how the mutagenesis and MD results fit in with the structural work, we have also revised this section and highlighted in red as following:

“MD simulations (Extended Data Fig. 5c and d) indicated that the larger side chain of T307L increased steric hindrance with substrate **1**, altering its binding and favoring the reaction at the lower energy barrier of C2'. Given the structural proximity of Y308M to T307L, we hypothesize that Y308M may similarly affect substrate binding.... These findings suggest that the residues F273, T307 and Y308 in SsBcmE play a crucial role in regulating the site selectivity.... further supporting the idea that steric hindrance mediated by these residues controls site selectivity in SsBcmE-catalysed hydroxylation.

Together, we propose that BcmE regulates site selectivity by constructing a specific hydrophobic cavity that accommodates the orientation of substrate **1**, employing a steric hindrance control

strategy similar to those used in chemical synthesis (Fig. 4a). For instance, mutating the bulky F273 to alanine reduces the steric hindrance on the leucine side of substrate **1**, positioning the preferred reactive site, C6, closer to the center and facilitating hydroxylation. This observation is consistent with the reactivity predicted in Fig. 2b.”

One minor but serious issue that still remains in spite of the provided response (see Reviewer 2, minor comments, bullet points 7 and 8) is that “hydroxylation” should be replaced with “hydrogen abstraction” in both the header of Fig. 2b and the figure caption of Fig. 2. Regarding bullet point 10 in the minor comments, not all r.m.s.d. values have been truncated to two decimal places (e.g., caption of Supplementary Fig. 7). The newly added sentence (lines 62-65): “... such like ... Proline” has serious grammatical issues.

R: We have taken the suggestion to revise all the issues that were mentioned above. And the newly added sentence was revised to “Several α KG-dependent amino acid hydroxylases can hydroxylate the same amino acid at different sites. For example, VioC, OrfP, CmnC, and EFEs catalyze the hydroxylation of L-Arg or its analogs, while P3H and P4H selectively oxidize the C–H bonds of proline.”

b

Calculated energy barriers for different types of hydrogen abstraction (kcal/mol)

site	BcmE	BcmC	BcmG
1'	9.9(11.8) [R(S)]	8.2(12.1) [R(S)]	9.3(11.7) [S(R)]
2'	6.4	5.1	--- (-OH)
3'	10.8(12.9) [S(R)]	9.4(10.4) [R(S)]	9.8(12.8) [R(S)]
5	8.7	6.9	5.3
5a	14.4	11.9	7.9
6	6.6(8.2) [R,R(R,S)]	8.7(10.0) [R,R(R,S)]	9.1(10.8) [R,S(R,R)]
7	12.5	--- (-OH)	--- (-OH)

Revised Fig.2. DFT-computed transition states and Gibbs free energies barrier (in kcal mol⁻¹) for the hydrogen abstraction reactions by a truncated catalytic-residue theozyme model.

Reviewer #3 (Remarks to the Author):

All issues have been addressed well, publication is recommended.

R: Thanks a lot!

Responses to Referees Letter

We sincerely appreciate the comments from the reviewers, which have been very helpful in enhancing the quality of our manuscript. We have taken the suggestion to add the corresponding reference 76 in line 105 of the revised manuscript.

Our point-by-point responses to the reviewers is included below in this file.

Remarks from the reviewers are shown in black. Our responses are shown in **blue**.

Reviewer #2 (Remarks to the Author):

The authors have generally addressed the previous concerns.

On lines 115-118 of the previous submission, the authors have added some references to research articles describing the importance of substrate positioning in determining reaction outcome in Fe/2OG enzymes. Please add Matthews et al. (2009) "Substrate positioning controls the partition between halogenation and hydroxylation in the aliphatic halogenase SyrB2." PNAS 106, 17723.

R: We thank the reviewer's suggestion very much and have added the corresponding reference into the revised manuscript at line 105:

76 Matthews, M. L. et al. Substrate positioning controls the partition between halogenation and hydroxylation in the aliphatic halogenase, SyrB2. *Proc. Natl. Acad. Sci. U.S.A.* **106**, 17723-17728 (2009).